# Loss of luminal lineage drives resistance to next-generation ERα antagonists in pretreated ER+ HER2− locally-advanced or metastatic breast cancer

Jackson Liang [1] ✉, Christy Ong[2], Kareem Heslop[2], Jane Guan[2], Vasumathi Kameswaran [3], Bence Daniel [3], Minyi Shi[3], Yuxin Liang[3], Jennifer M. Giltnane [4], Junko Aimi[1], Ching-Wei Chang[5], Mary R. Gates[6], Jennifer Eng-Wong[6], Pablo Perez-Moreno[7], Komal L. Jhaveri [8], Nicholas C. Turner [9], Elgene Lim [10], Ciara Metcalfe[2] & Heather M. Moore[1]

Next-generation selective estrogen receptor-α (ERα) antagonist/degraders (SERDs) are being developed for ER-positive breast cancer (ER+ BC), with intentions of improving outcomes for patients. In recent clinical trials of metastatic ER+ BC, next-generation SERDs demonstrated clinical activity, and elacestrant received an approval for advanced *ESR1*-mutant disease. However, responses to these drugs were highly heterogeneous: across trials and independent of *ESR1* status, 30–50% of patients progressed by their first follow-up scan while other patients sustained benefit for 2 years or more. Here, we interrogate the basis for heterogeneous responses by comparing biopsies from non-responding patients (NR; progression-free survival <2 months) and responding patients (Resp; PFS ≥ 2 months) who received the next-generation SERD giredestrant. While Resp tumors maintain high dependency on ERα signaling, NR tumors exhibit loss of luminal lineage identity and by extension, ERα dependence. NR tumors instead up-regulate multiple ERα-independent proliferative pathways, such as EGFR/MAPK and Hippo/TEAD, which may represent targetable dependencies in NR disease. Modeling resistance and lineage plasticity in vitro, we find that giredestrant-resistant ER+ BC cell lines exhibit profound shifts in chromatin accessibility, with the transcription factors, FOXA1 and FOXM1, implicated in gene expression of NR-upregulated proliferative pathways.

Breast cancer is the most frequently diagnosed cancer worldwide and a major contributor of mortality[1]. Approximately 70% of breast cancers belong to the estrogen receptor-positive (ER+) subtype, which is driven by estrogen receptor-α (ERα). Endocrine therapies that inhibit the ERα signaling pathway are the mainstay treatment strategy for this subtype.

ER+ breast cancers are commonly treated with aromatase inhibitors (AIs), which block estrogen synthesis. However, prominent use of AIs have selected for *ESR1* (encodes ERα) hotspot mutations, whereby mutant ERα adopts a constitutively active conformation to enable estrogen-independent activation[2–5]. Later lines of therapy include ERα

therapeutic ligands such as fulvestrant, the prototypical selective ERα antagonist/degrader (SERD), which binds ERα to antagonize, immobilize and degrade it[6]. Unfortunately, poor bioavailability[7] limits the efficacy of fulvestrant in both *ESR1*-mutant tumors and other metastatic disease.

Deficiencies associated with existing endocrine therapies inspired widespread development of next generation ERα-targeted therapies[8–13], with the intention of improving benefit in all patients including those harboring *ESR1* mutations. In the past several years, multiple investigational oral SERDs have been evaluated in clinical trials, including giredestrant[9,10,14], elascestrant[11], camizestrant[13], amcenestrant[15], and imlunestrant[16]. These molecules exhibit on-target activity against the ERα pathway in ER+ breast cancer. Next-generation SERDs showed superior monotherapy activity against *ESR1*-mutant disease compared to standard-of-care endocrine therapies in the advanced/metastatic setting. However, patients had profoundly heterogeneous responses to these drugs. Across all clinical trials—acelERA[17], EMERALD[11], SERENA-2[13], AMEERA-3[18], and EMBER-3[19] (NCT04576455, NCT03778931, NCT04214288, NCT04059484, and NCT04975308)—and across experimental and standard-of-care arms, 30–50% of patients had progressed by their first on-treatment scan, while others sustained benefit up to 2+ years. These differential responses were observed in both *ESR1*-mutant cases and the intent-to-treat (ITT) population.

We therefore explored the mechanistic basis for heterogeneous response to SERDs in metastatic ER+ disease by profiling patients with advanced ER+/HER2- breast cancer enrolled in a phase-1a/b study of giredestrant. Here, we show that poor responses to giredestrant were associated with lineage plastic disease and elucidate the molecular underpinnings of lineage-plastic ER+ breast cancer.

## Results
### Retrospective biomarker analysis on a phase-1a/b giredestrant cohort

We performed exploratory biomarker analyses on biopsies from patients with advanced/metastatic ER+ breast cancer (mBC) enrolled in a phase-1a/b giredestrant dose escalation and expansion clinical trial (GO39932, NCT03332797; Supplementary Fig. 1a) [14]. Eligible patients had confirmed ER+/HER2- locally-advanced or metastatic breast cancer, received ≤ 2 prior lines of therapy in this setting, and remained on adjuvant endocrine therapy for ≥ 24 months and/or derived a clinical benefit from endocrine therapy in the advanced/metastatic setting. Patients received giredestrant as a monotherapy (10, 30, or 90/100 mg doses) or in combination with 125 mg palbociclib, a CDK4/6 inhibitor (CDK4/6i). Pre-menopausal patients were additionally administered luteinizing hormone-releasing hormone (LHRH) agonists to suppress ovarian function.

Patients provided liquid and tumor FFPE biopsies for retrospective biomarker analyses (Supplementary Fig. 1b). Biopsied patients were generally representative of the study population (Supplementary Fig. 1c). We assessed molecular response to giredestrant via Ki-67 immunohistochemistry (IHC) on *n* = 29 patients with matching baseline and on-treatment biopsies (C2D8; cycle-2, day-8). Monotherapy giredestrant reduced Ki67 in most cases across dose levels (Supplementary Fig. 1d). All patients who received giredestrant plus palbociclib experienced Ki67 reduction and better responses by RECIST criteria (Supplementary Fig. 1e). Nearly all tumors were either LumA or LumB by PAM50 subtyping (Supplementary Fig. 1f–g). 37% and 31% of patients harbored *ESR1* and *PIK3CA* mutations, respectively, as measured by digital droplet PCR (ddPCR) and the comprehensive genomic profiling panel, FoundationACT (F-ACT; Supplementary Fig. 2a–f). However, mutation calls from tumor RNA-seq reads had poor agreement with ddPCR and F-ACT, consistent with known discrepancies between tissue versus liquid-based approaches[20]

(Supplementary Fig. 2g, h). This cohort's broader genomic landscape was similar to previous reports in this setting[21,22] (Supplementary Fig. 2c).

### Benefit on giredestrant is associated with ERα-dependent luminal disease, in both *ESR1*-mutant and NMD tumors

To interrogate the differences between giredestrant-responsive versus intrinsically-resistant disease, we grouped patients by progression-free survival. Patients who progressed by the time of the first scan (PFS < 2 months) were categorized as non-responders (NR), while those with PFS ≥ 2 months were defined as responders (Resp; Fig. 1a). These groups aligned with RECIST responses, with all progressive disease (PD) occurring in NR patients (Fig. 1a). We performed univariate Cox proportional hazards modeling on key clinical features—demographics, disease characteristics, and treatment history—to assess their influence on PFS in *n* = 181 patients (Supplementary Fig. 3a). Only CDK4/6i-related features were associated with PFS (Supplementary Fig. 3a-b). Prior CDK4/6i exposure was associated with significantly worse PFS, while patients enrolled in the palbociclib combination arm had improved PFS, consistent with the prior PALOMA-3 results[23]. These two effects are difficult to differentiate, since most of the patients in the palbociclib combination arm had no prior CDK4/6i exposure (Supplementary Fig. 3c). *ESR1* status was not associated with PFS (Supplementary Fig. 3a, b, d), consistent with other studies in which *ESR1*-mutant cases treated with next-generation SERDs achieved similar median PFS compared to the ITT population[11,13,17]. Likewise, *ESR1*-mutant and NMD (no mutation detected) cases had similar RECIST responses and proportions of NRs versus responders (Supplementary Fig. 3e).

To identify the molecular cause of differential response, we unbiasedly compared gene expression between baseline and on-treatment biopsies in responders and non-responders (Fig. 1b). Giredestrant did not significantly alter gene expression in NR tumors. By contrast, giredestrant repressed a program of 144 genes in Resp tumors (log$_2$-fold change ≤ −1, FDR < 0.05). These were predominantly comprised of cell cycle genes, such as *MKI67*, *CDC25A*, and *CCNA2*. Unbiased HALLMARK gene set enrichment analysis revealed that suppressed genes were associated with either estrogen response or cell cycle (Fig. 1c). Thus, giredestrant inhibits ERα- and proliferation-associated genes in Resp, but not NR tumors.

We explored the basis for differential ERα inhibition by examining key biomarkers. Given the strong influence of CDK4/6i on response (Supplementary Fig. 3a, b), we restricted our initial analyses to patients who received giredestrant monotherapy, paralleling the recent studies in late-line mBC. Among monotherapy-treated patients with baseline biopsies (*n* = 38), NR cases had a higher median Ki67 %-positivity at baseline (*p* = 0.06; Fig. 1d). Resp patients had significantly higher levels of baseline ERα protein compared to NR (Fig. 1d). We assessed giredestrant-ERα engagement by comparing ERα IHC H-scores at baseline and on-treatment for patients with paired biopsies (Supplementary Fig. 4a). Giredestrant reduced ERα to similar extents in NR and Resp tumors, consistent with on-target activity in both groups. Responders also exhibited elevated baseline levels of PgR (progesterone receptor), which were significantly reduced on-treatment (Fig. 1d, Supplementary Fig. 4a). Expression of *ESR1* and *PGR* was likewise elevated in Resp patients compared to NR (Supplementary Fig. 4b). Since *PGR* is an ERα-target gene, we hypothesized that NR and Resp tumors exhibit differential ERα pathway activation.

We examined ERα activity by evaluating established estrogen-induced and estrogen-repressed genes from preclinical models[6]. Since ERα-induced genes were more impacted by giredestrant exposure (Supplementary Fig. 4c, d), we used this signature to infer ERα activity. At baseline, responders exhibited higher ERα activity than NRs (Fig. 1d); this activity was significantly reduced by giredestrant

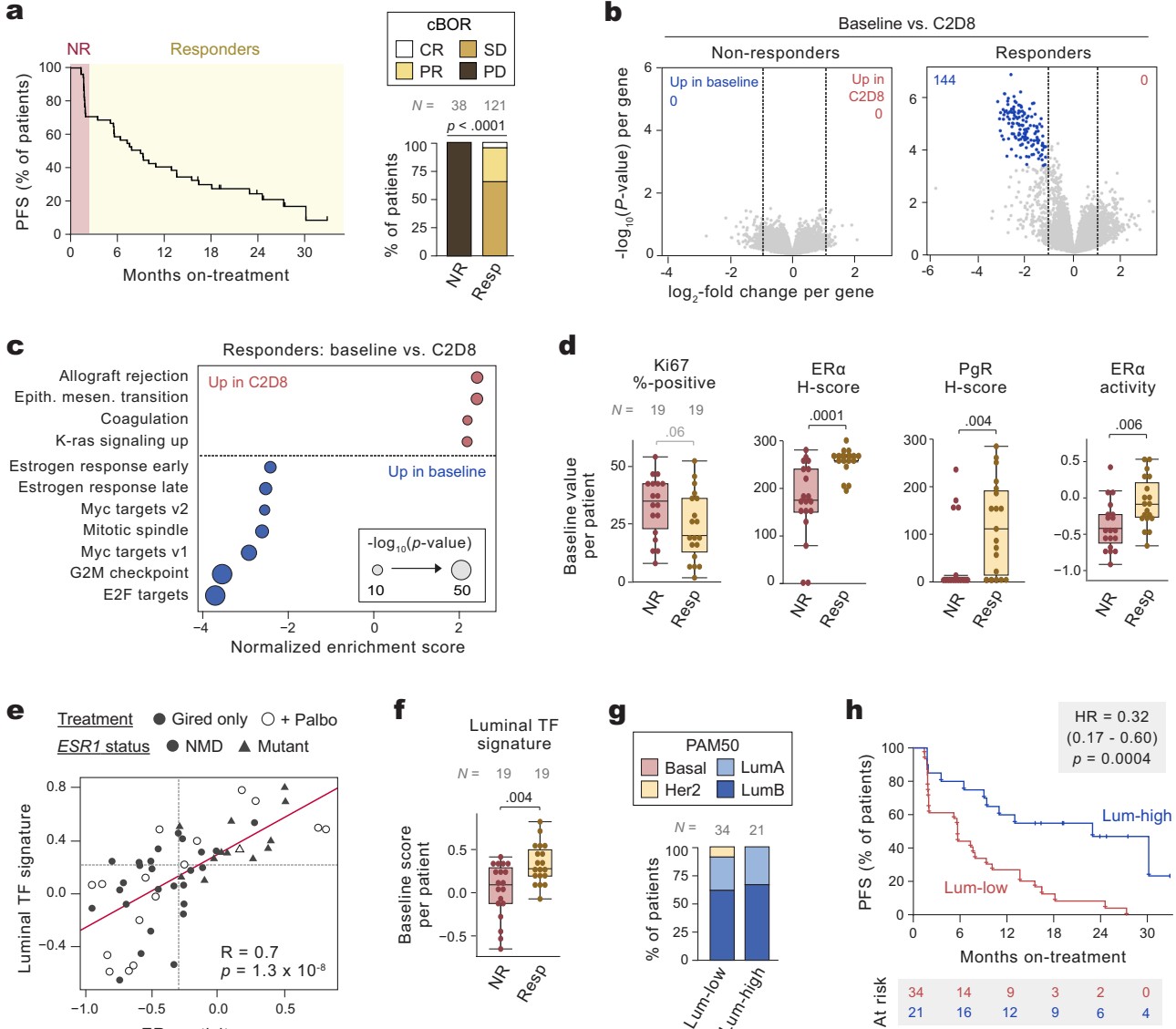

**Fig. 1 | Giredestrant elicits therapeutic response in late-line metastatic tumors with high ERα activity and luminal lineage program. a** Progression-free survival (PFS) for patients with evaluable tumor biopsies. Patients are stratified by PFS into non-responders (NR; PFS < 2 months) or responders (Resp; PFS ≥ 2 months). (Left) Dashes denote censored events. (Right) confirmed best overall response (cBOR) by RECIST; CR: complete response, PR: partial response, SD: stable disease, PD: progressive disease. *P*-values via two-sided χ²-test. **b** Differential gene expression of C2D8 vs. baseline RNA-seq for NR or Resp patients. Each point is one gene. Colored points represent genes with FDR < 0.05; statistics via the voom-limma method. Dotted lines denote log₂-fold change cutoffs of ±1. **c** For Resp cases, HALLMARK gene set enrichment analysis; each point is one gene signature. Signatures achieving a normalized enrichment score (NES) of ± 2 and FDR < 0.05 are shown. *P*-values via the 'fgsea' (two-sided) method. **d** Key biomarkers for baseline biopsies by response. **e** Comparison of ERα activity and luminal TF scores in baseline

biopsies. Biopsies with greater than median ERα activity and luminal TF scores (dotted lines) are categorized as 'Lum-high'; all others are categorized as 'Lum-low'. R and *p*-value via two-sided Pearson's correlation method. *ESR1* mutational status was inferred by RNA-seq reads (see "Methods"). **f** Luminal TF scores per baseline biopsy by response. **g** PAM50 subtyping at baseline by luminal status; data represent individual biopsies categorized by luminal score. **h** Comparison of PFS in Lum-high vs. Lum-low patients, as defined in (**e**); *p*-value and hazard ratio via log-rank test. In (**b**, **c**), *n* = 10 NR and 17 Resp pairs of biopsies (baseline and on-treatment, C2D8) evaluated. In (**d**, **f**), comparisons are restricted to patients who received monotherapy giredestrant. Boxes are 25th percentile, median, and 75th percentile, whiskers are maximum/minimum values within 1.5-times the interquartile range (IQR). *P*-values were calculated by unpaired two-sided Mann-Whitney *U*-test. In (**d**–**f**), each point is one baseline biopsy; *n* = 19, 19, and 16 for monotherapy NR, monotherapy Resp, and combo Resp, respectively. Source data is available.

treatment (Supplementary Fig. 4a). By contrast, NRs had low baseline ERα activity comparable to that of giredestrant-treated responders. In a paired baseline and on-treatment analysis, tumors with high baseline ERα activity (categorized by the median value) experienced deep and concomitant suppression of both ERα activity and the E2F proliferation gene signature on-treatment (Supplementary Fig. 4e). In summary, Resp tumors maintain high levels of ERα protein and target gene expression, while NR tumors exhibit reduced ERα protein and low

target expression. These data suggest monotherapy Resp cases are highly dependent on ERα and its inhibition via giredestrant drives benefit, while NR cases have lost this dependency and do not respond to giredestrant.

Next, we sought to better understand the heterogeneity in ERα dependence among pretreated ER⁺ mBC. Loss of hormone receptor dependence has similarly been observed in prostate cancer, in which early disease is driven by androgen receptor (AR). As prostate tumors

are exposed to prolonged AR pathway inhibition, a subset of tumors undergo a lineage switch from the AR$^+$ luminal state to the AR$^-$ neuroendocrine state and become insensitive to AR-targeted agents[24,25]. By analogy, early ER$^+$ tumors inherit lineage identity (and thus ERα dependency) from the luminal breast cells. We therefore asked whether differences in ERα activity/dependence could be explained by shifts in the luminal lineage. To address this question, we leveraged our previously-generated breast cancer lineage-specific transcription factor (TF) signatures[26]. These signatures were defined using TCGA data as the set of TFs respectively enriched in luminal breast cancer (LumA, LumB) versus basal or triple-negative breast cancer. As expected, tumor ERα activity has a strong positive correlation with the luminal TF signature (Fig. 1e).

We asked whether the luminal lineage-specific TF signature was associated with response. Indeed, Resp patients had significantly higher luminal TF scores at baseline than NR patients (Fig. 1f). Despite the shifts in lineage-specific TF expression, PAM50 subtypes remained similar between luminal-high vs. -low tumors (luminal-high tumors had greater than median ERα activity and luminal TF scores; Fig. 1g). Similarly, all tumors retained baseline ERα-positivity of ≥1%. This suggests that although NR tumors have lost some lineage features associated with luminal breast cancer, they have not undergone a wholesale switch of breast cancer subtypes. Finally, we evaluated whether luminal disease was associated with outcome. Independent of study arm, patients with luminal-high tumors had significantly improved PFS compared to luminal-low cases, with median PFSs of 22 months versus 6 months, respectively (Fig. 1h). Taken together, in the context of giredestrant monotherapy, Resp tumors maintain a strong luminal lineage identity and ERα activity/dependence, whereas NR tumors lose some luminal features and become ERα-independent.

We next assessed patients receiving giredestrant in combination with palbociclib. Since patients receiving the palbociclib combination were predominantly responders (Supplementary Fig. 1e), we compared Resp cases receiving monotherapy (Gired) to those receiving the giredestrant-palbociclib combination (Combo). At baseline, Combo Resp tumors exhibited significantly greater variance in ERα protein levels and luminal TF score compared to monotherapy (Supplementary Fig. 4f). ~70% of Combo Resp tumors exhibited comparable levels as Gired Resp tumors, and the remaining 30% had lower ERα H-scores, ERα activity, and luminal TF score. Likewise, an unbiased gene set enrichment analysis confirms that estrogen response genes were more enriched in Gired Resp tumors (Supplementary Fig. 4g). Conversely, NR-enriched pathways (discussed below) were enriched in Combo tumors (Supplementary Fig. 4f, g). These data suggest that in the context of a CDK4/6i combination, tumors with low ERα dependence (resembling NR tumors in the monotherapy arm) may still experience benefit since CDK4/6 inhibitors target an orthogonal dependency.

## Non-responding patients are enriched for putative resistance pathways

We hypothesized that as tumors lose their ERα dependency and luminal identity, they must up-regulate orthogonal signaling pathways to sustain tumor growth. To identify these pathways, we compared luminal-high versus luminal-low tumors by gene set enrichment analysis (Supplementary Fig. 5). As expected, luminal-high tumors were significantly enriched for estrogen-related gene expression. By contrast, luminal-low tumors were significantly enriched for multiple proliferation pathways, including EGFR/MAPK, Hippo/TEAD, Wnt, and others. Across collections, approximately half of all top-enriched gene signatures in luminal-low tumors were associated with immune function (e.g. T-cell and B-cell receptor signaling, interleukin signaling, interferon, PD-1). Of particular interest was the enrichment of inflammatory JAK/STAT, which has been linked to lineage plasticity in prostate cancer[27].

We asked if activity in these pathways was associated with response in our cohort. Among monotherapy giredestrant-treated patients, NR patients had significantly higher gene expression scores for MAPK, TEAD, Wnt, and IL2/STAT5 pathways compared to responders (Fig. 2a). Of these, MAPK/RTK (receptor tyrosine kinase) signaling and alterations have previously been linked to endocrine resistance[22,28,29]. Elevation of specific pathways is not easily explained by genomic alterations. While multiple genes associated with orthogonal pathways exhibit increased mutational frequency in NR (e.g. ERBB2), most patients with elevated MAPK signaling did not harbor genomic alterations in that pathway (Fig. 2b). These orthogonal pathways may provide a mechanistic basis for ERα-independent disease progression in NR patients.

To further compare non-responders and responders, we performed generalized linear modeling on expression data, with treatment arm and PAM50 as co-variates. This approach identified genes which were significantly associated with either NR or Resp tumors, measured by log$_2$-odds ratio (Fig. 2c). We identified 1,023 and 1,254 genes associated with NR and Resp tumors, respectively. NR-genes included key regulators or downstream targets of MAPK/RTK signaling (e.g. EGFR, MAPK1, MAPK6), TEAD signaling (TEAD4, ANLN, F3), and Wnt signaling (FZD4, FZD5, FZD8), which unbiasedly confirms the elevation of these pathways in NR tumors. Evaluating the expression of these genes, NR and Resp cases exhibited significant elevation of their respective associated genes (Fig. 2d). Among responders, both monotherapy and Combo patients had significantly elevated Resp-gene expression and significantly depleted NR-gene expression compared to NR patients. Combo Resp cases had intermediate expression of these genes, with significantly elevated NR-genes and decreased Resp-genes relative to monotherapy Resp tumors (Fig. 2d). The gene scores successfully separated NR and Resp tumors, except for a small number of Combo Resp tumors which overlapped with the NRs (Fig. 2e).

Since NR tumors lose expression of luminal TFs, we asked what TFs may exhibit increased activity in this context. We performed transcription factor enrichment analysis[30] to identify the TFs known to regulate NR- and Resp-genes, respectively. As expected, ERα (ESR1) was significantly associated with Resp-genes (Fig. 2f). By contrast, NR-genes were significantly associated with transcription factors such as TEAD4 and TCF4, consistent with the observed elevation of Hippo/TEAD and Wnt pathways in NR tumors. Surprisingly, we observed significant enrichment of FOXM1, a transcription factor which is known to both cooperate with ERα to regulate proliferation in ER$^+$ breast cancer cells[31,32] and drive ERα-independent proliferation in basal/triple-negative breast cancer[33,34] (further discussed below). Taken together, these analyses suggest that while NR tumors exhibit increased transcription factor and pathway activity in multiple proliferative pathways.

## Pre-clinical models of giredestrant resistance exhibit loss of luminal identity and gain of NR-associated pathways

To further interrogate NR-associated molecular features, we developed in vitro models of giredestrant resistance in MCF7 and T47D cells. Each model was cultured long-term in 100 nM giredestrant and acquired giredestrant resistance over the course of 4 months (GiredR; Fig. 3a). As expected, giredestrant immediately reduced ERα protein. Over 4 months of passages, both cell lines further downregulated ERα protein (Fig. 3a). While parental cells were sensitive to giredestrant, fulvestrant, and tamoxifen (a SERM; selective ERα modulator), GiredR cells exhibited resistance to multiple ERα therapeutic ligands at concentrations of 1 μM or more (Fig. 3b).

To assess the evolving transcriptome as cells acquire resistance, we compared (1) parental cells, (2) cells at 2 months on-treatment, which retained some ERα protein and had not acquired full resistance, and (3) GiredR cells by bulk RNA-seq. GiredR cells exhibited profound

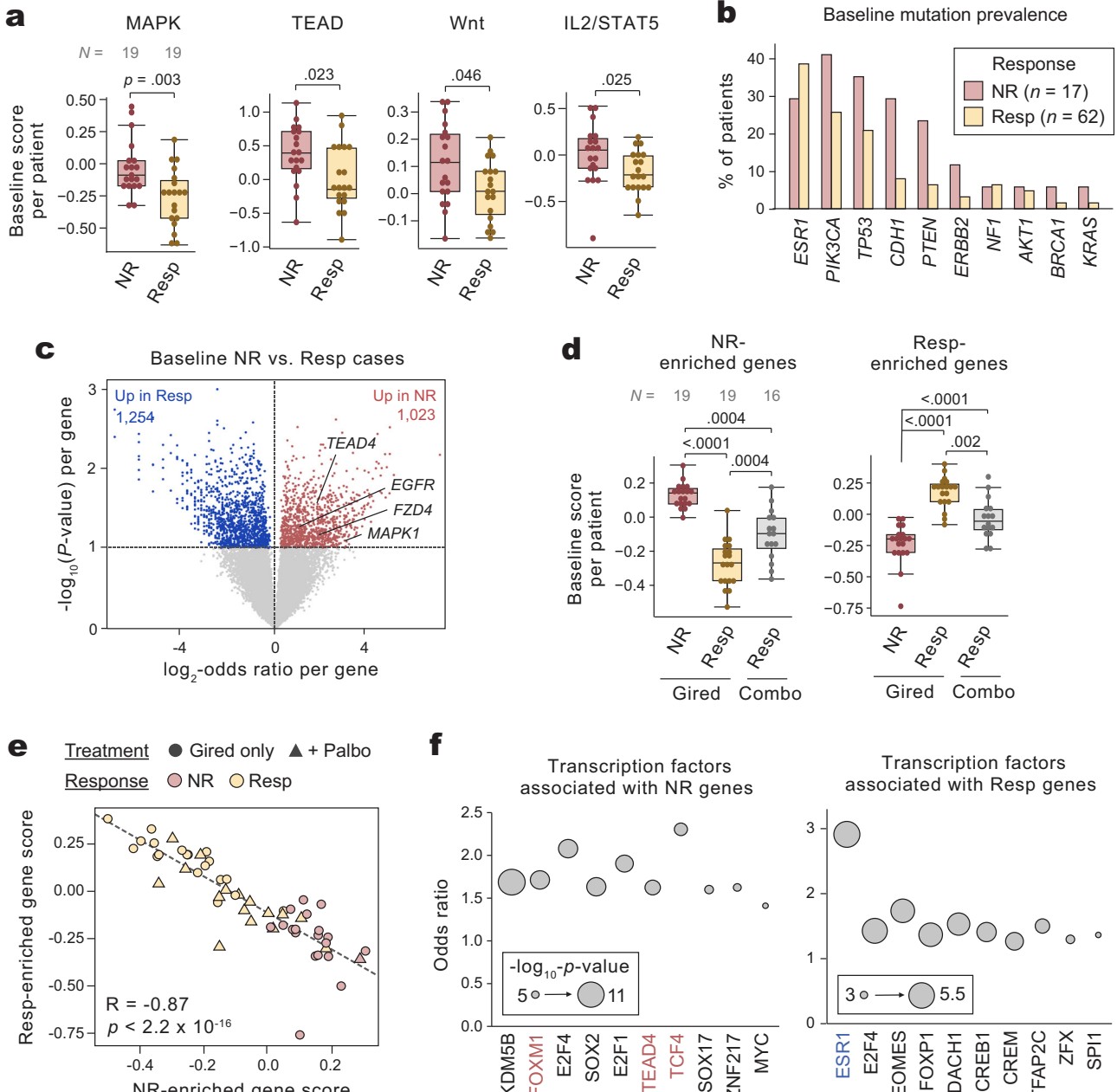

**Fig. 2 | Non-responding tumors exhibit increased pathway and transcription factor activity in orthogonal oncogenic pathways. a** Gene signature scores (from Lum-low enriched pathways; see Supplementary Fig. 5) by response, for patients receiving monotherapy giredestrant. **b** Prevalence of short variants in cancer-associated genes by response; mutations via F-ACT. Both monotherapy gir-edestrant and palbociclib-combination patients are included. (**c**) Genes associated with either NR or Resp baseline biopsies, evaluated by generalized linear modeling with arm (monotherapy vs. combination) and PAM50 subtype as covariates. Each point is one gene, enriched in either response group; p-values by two-sided Wald' test. **d** Gene signature scores computed for 1023 NR- or 1254 Resp-enriched genes (defined in [**c**]) per baseline biopsy. Resp patients are stratified by treatment arm.

**e** Comparison of NR- and Resp-enriched gene scores per baseline biopsy (n = 55 biopsies); R and p-value via two-sided Pearson's correlation method. **f** Transcription factor (TF) enrichment analysis for NR- and Resp-enriched genes; per gene set, the top 10 TFs computed based on literature-reported ChIP-seq data and FDR < 0.05 are shown. Each point represents the odds ratio for one TF; p-values via one-sided Fisher's exact test. In (**a**, **d**), boxes are 25th percentile, median, and 75th percentile, whiskers are maximum/minimum values within 1.5-times the IQR. P-values by unpaired two-sided Mann-Whitney U-test. In (**a**, **d**, **e**), each point is one baseline biopsy; n = 19, 19, and 16 for monotherapy NR, monotherapy Resp, and combo Resp, respectively. Source data is available.

transcriptomic changes compared to the respective parental cells (Supplementary Fig. 6a). For most differentially-expressed genes, changes could already be observed at 2 months (Supplementary Fig. 6b). We asked whether GiredR cells recapitulated the features of NR patients in pretreated ER⁺ mBC. ERα activity was markedly reduced within 2 months of treatment, consistent with on-target giredestrant activity (Fig. 3c) and similar to prior observations in fulvestrant-

resistant cell lines[35]. Luminal TF expression was also reduced in GiredR cells (Fig. 3c). However, this reduction occurred later than the acute effect on ERα activity, consistent with prior findings that loss of luminal TFs requires prolonged endocrine suppression[26]. These data also suggest ERα activity and luminal identity can be independently measured; the luminal TF signature remains high at 2 months despite decreased ERα activity. NR-enriched pathways were enriched in GiredR

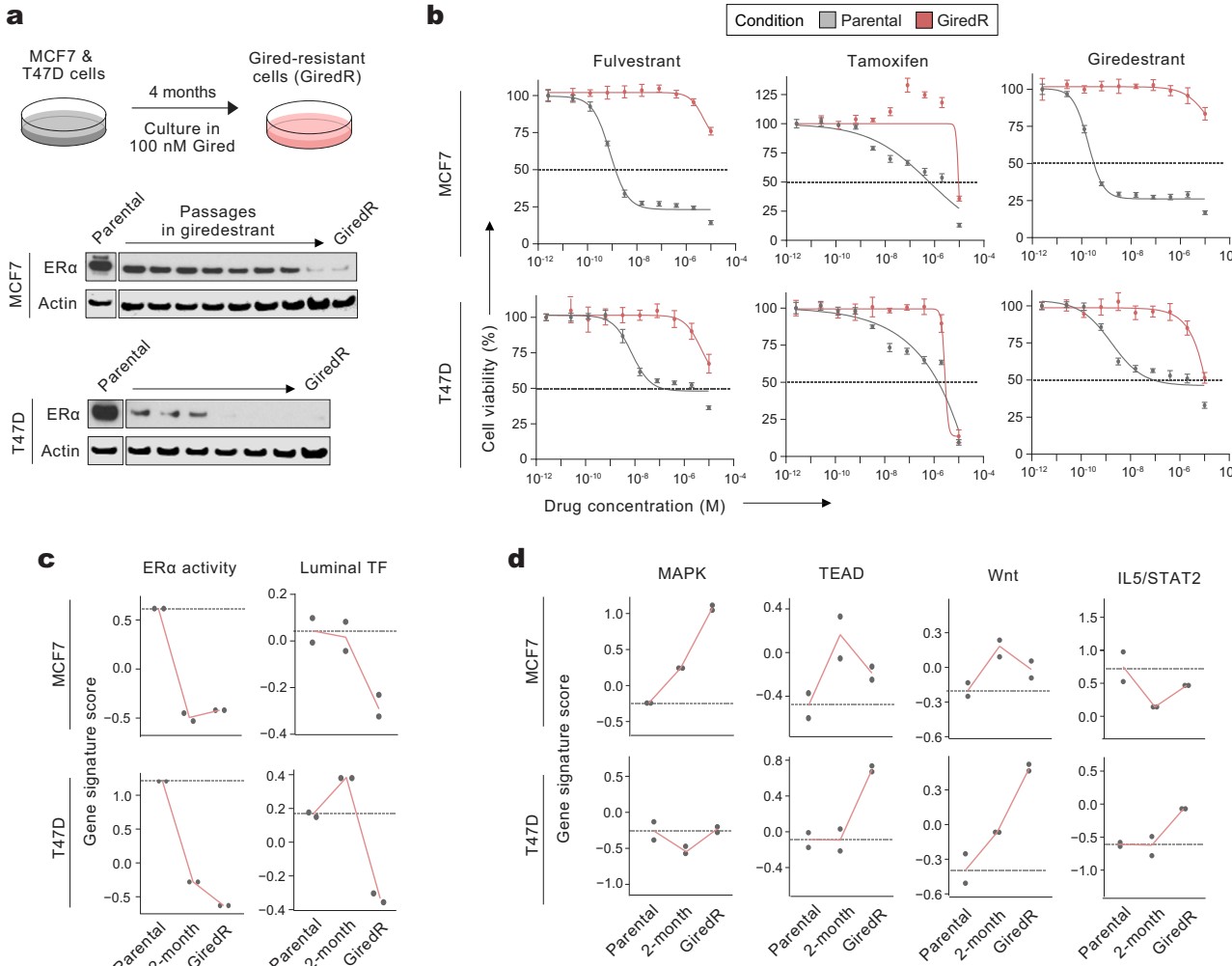

**Fig. 3 | Pre-clinical models of giredestrant resistance recapitulate features of metastatic NR patients. a** Experimental approach for generating giredestrant-resistant (GiredR) MCF7 and T47D cell lines. Cells were cultured in 100 nM giredestrant for 4 months, until cells developed drug resistance. For Western blots, protein samples between parental and GiredR samples were from cells collected at each passage during the 4-month giredestrant treatment period, starting from left to right. Experiment was performed once. **b** Cell viability for anti-ERα therapeutic ligands for parental and GiredR cell lines; $n = 4$ biological replicates per cell line. Cell viability was measured using the CellTiterGlo assay; dotted lines: IC50 value (half-maximal inhibitory concentration) per drug. Values are mean ± error. **c**, **d** Gene signature scores via RNA-seq per sample ($n = 2$ biological replicates per condition) for ERα activity and luminal TF signatures (**c**) or NR-associated signaling pathways (**d**). Each point is the calculated score for one sample; red lines connect mean values across conditions. Dotted lines: mean value in respective parental cells. Source data is available.

cells, although differences between the two models were observed (Fig. 3d). GiredR-MCF7 cells exhibited a profound increase in MAPK pathway activity, and modest increases in TEAD and Wnt activity. By contrast, Gired-T47D cells did not have elevated MAPK activity, but markedly induced TEAD and Wnt activity and upregulated IL2/STAT5 signaling. Unbiased gene set enrichment analysis corroborated these findings (Supplementary Fig. 6c).

We also profiled mutations acquired by GiredR cells. By whole exome sequencing, a limited number of cancer-associated short variants were acquired in MCF7 and T47D GiredR cells. In GiredR-MCF7, variants in *FGFR3*, *HOXD13*, and *RABEP1* were detected; the *FGFR3* variant was V117I, which has no known association with disease. In GiredR-T47D, *SETD2*, *HSP90AA1*, *KBTBD4*, *TLE1*, and *KDM6A* were mutated. Likewise, few copy number variations (CNV) were detected. Evaluating CNV between GiredR and parental with log$_2$-copy ratio cutoffs of 1 (amplified) or −1 (loss/deletions), no cancer-associated genes were amplified in either GiredR lines. Of the two GiredR cell lines, only MCF7-GiredR had deletions—specifically, 19 cancer-associated genes on chromosome X, of which only *UBE2A*, *XIAP*, *STAG2*, *BCORL1*, *PHF6*, *RPL10*, and *ATP6AP1* were expressed in MCF7 cells. Notably, overexpression of *ATP6AP1* was previously linked to tamoxifen resistance[36], but its deletion here is unlikely to explain resistance in MCF7-GiredR.

As a complementary approach, we generated MCF7 and T47D cell lines, which were resistant to GDC-0810[37], a therapeutic ligand exhibiting pharmacological features of both SERDs and SERMs (SERD-SERM hybrid). GDC-0810 resistant cells (0810 R) were established by culturing parental cells with increasing concentrations of GDC-0810 over 6 months (Supplementary Fig. 7a). Two clones per cell line were analyzed. As with GiredR cells, the 0810 R cells were cross-resistant to fulvestrant and tamoxifen at concentrations of 1 μM or more. Luminal TFs were repressed across 0810 R clones (Supplementary Fig. 7b). In both 0810 R models, transcriptional profiling revealed the loss of ERα activity and enrichment of orthogonal pathways (Supplementary Fig. 7c, d), and proteomic profiling via reverse phase protein array (RPPA) revealed decreased ERα protein (Supplementary Fig. 7e). Protein levels of EGFR and MAPK pathway components were increased in 0810R-MCF7, but not in 0810R-T47D cells. Together, these data

demonstrate that the same resistance-associated features emerge across two different classes of anti-ERα therapeutic ligands: full SERDs (giredestrant) and SERD-SERM hybrids (GDC-0810).

## ERα inhibition promotes increased chromatin accessibility at ERα-independent sites associated with FOXA1 and FOXM1

We asked whether epigenetic alterations may promote ERα independence upon prolonged ERα inhibition. We performed bulk ATAC sequencing to evaluate the chromatin accessibility landscape in parental cells, 2-month-treated cells, and GiredR cells. For both models, GiredR cells exhibited profound shifts in chromatin accessibility compared to their parental counterparts (Fig. 4a). GiredR cells had altered accessibility at 47,380 and 22,446 sites in MCF7 and T47D, respectively (34% and 17% shifts in accessibility). In both models, the number of chromatin sites gained in GiredR cells was 4 to 6-fold higher than the number of sites lost. Loss of GiredR-repressed sites (down in GiredR) occurred prior to the elevation of GiredR-induced sites (up in GiredR; Fig. 4b, c). Combined analyses across models yielded similar results (Supplementary Fig. 8a–c). To probe the functional importance of GiredR-repressed and induced chromatin sites, we performed motif enrichment analysis on each. The ERα (ESR1) motif was significantly enriched among repressed sites (Fig. 4d), indicating that ERα binding sites become inaccessible in GiredR cells. By contrast, GiredR-induced sites were enriched for multiple Forkhead (FOX)-family members and TEAD (Fig. 4d, Supplementary Fig. 8d). Across induced sites, the top enriched motifs were FOXM1 and FOXA1 (Fig. 4d); although other FOX-family motifs were enriched, these factors are not expressed in either cell line. The enrichment of FOXM1-associated chromatin sites parallels the association between NR-specific genes and FOXM1 transcriptional activity in NR patients (Fig. 2f). However, the enrichment of FOXA1 sites was surprising given its role as a master regulator of the luminal lineage in both healthy breast and ER+ BC.

FOXA1 normally acts as a pioneer factor for ERα, enabling accessibility at key ERα-target and luminal genes[38,39]. We assessed the co-occurrence of ESR1 and FOXA1 motifs among GiredR-repressed and induced sites. Evaluating all sites containing ERα and/or FOXA1 motifs, 70% of GiredR-repressed sites contained an ERα motif without a co-occurring FOXA1 motif (Supplementary Fig. 8e). Conversely, 70% of GiredR-induced sites are represented by FOXA1 alone. A similar pattern is observed when comparing ERα to FOXM1 motifs among differential GiredR sites, with FOXM1 motifs occurring distinctly from FOXA1 in half of relevant sites (Supplementary Fig. 8e, f). We characterized ERα-independent FOXA1 sites in GiredR cells. Compared to parental FOXA1 sites, GiredR FOXA1 sites were enriched for motifs of MAPK-associated factors (MEF2B, MEF2D), TEAD factors (TEAD1, TEAD4) and STATs (STAT5B, STAT4; Fig. 4e). Some model-specific motifs were observed, such as the enrichment of AR (ANDR) motifs in T47D (Supplementary Fig. 8g). Sites containing FOXM1 motifs were likewise enriched for these motifs (Fig. 4e, Supplementary Fig. 8g).

To confirm differential FOXA1 and FOXM1 binding in parental versus GiredR cells, we performed ChIP-seq (Supplementary Fig. 9a). For FOXA1, 135,218 and 144,854 ChIP sites were captured in MCF7 and T47D, respectively, with 54% of sites containing the FOXA1 motif. FOXA1 exhibited differential chromatin binding at 40–50,000 sites (Fig. 4f, g, Supplementary Fig. 9b), approximately half of which were shared between models (Supplementary Fig. 9c). In both GiredR models, FOXA1 was bound to the promoters of NR pathway genes, such as MAPK (MAP4K3, KRAS, DUSP10), TEAD (YAP1, ANLN, TAGLN), and Wnt (TCF4, CTNNB1, FZD4). Motif enrichment analyses of GiredR-enriched FOXA1 ChIP sites identified an enrichment of related motifs, such as TEAD1 and TEAD4 (Supplementary Fig. 9d). For FOXM1, 12,724 and 13,309 ChIP sites were respectively captured, consistent with a previous report[40]; 23% contained FOXM1 motifs. In GiredR cells, FOXM1 had elevated binding at 823 and 3,222 sites in the respective

models (Supplementary Fig. 9e-g). Although analysis identified parental-enriched FOXM1 sites, signal at these sites was weak and likely not meaningful. FOXM1 was bound to a higher frequency of promoter regions compared to FOXA1 (Supplementary Fig. 9h), was bound to promoters of key genes such as YAP1, and had enrichment of NR pathway-associated motifs such as TEAD1/4 (Supplementary Fig. 9i).

To test the functional importance of GiredR-induced chromatin sites, we identified the genes within 3 kb of chromatin sites in which FOXA1/FOXM1 motifs co-occurred with MEF2, TEAD, and STAT motifs (Supplementary Fig. 10a). Evaluating expression of these genes over treatment, we observed significant transcriptional induction in 2 month-treated and GiredR cells (Fig. 4h, Supplementary Fig. 10b). Compared to parental cells, genes proximal to FOXA1 + MEF2 sites were significantly elevated at 2 months and further elevated in GiredR (Fig. 4h). Likewise, genes proximal to FOXA1 + TEAD1/4 sites and FOXA1 + STAT4/5B sites were significantly elevated on-treatment. Similar expression trends were observed for genes near sites where FOXM1 co-occurred with these motifs (Supplementary Fig. 10b). Together, the data suggest that FOXA1/FOXM1 act at ERα-independent chromatin sites, which induce gene expression in pathways associated with NR disease.

## NR-associated pathways may be targetable in the context of giredestrant resistance

We sought to identify therapeutic vulnerabilities in the context of giredestrant resistance. Despite FOXM1's involvement in giredestrant resistance, GiredR cells were insensitive to the siRNA-mediated knockdown of FOXM1 (Supplementary Fig. 10c, d). We next considered the MAPK and TEAD pathways, given their upregulation in both NR patients and GiredR models. To broadly assess whether these pathways were associated with giredestrant resistance across preclinical models, we leveraged relative growth rates in giredestrant for 14 ER+/HER2- breast cancer cell lines using the GR method[41,42]. A relative growth rate of 1.0 denotes complete resistance (the cell line grows as well in giredestrant as in the absence of drug), whereas a relative growth rate of 0.0 denotes complete growth inhibition in the presence of giredestrant. We examined the association between growth rate in giredestrant and ERα, MAPK, and TEAD activity across the 14 cell line models. ERα activity exhibits a strong inverse correlation with the models' ability to grow in the presence of giredestrant, likely because it reflects each model's degree of ERα dependence (Supplementary Fig. 10e). Conversely, both MAPK and TEAD activities across cell lines were positively correlated with their ability to grow in the presence of giredestrant. The MAPK observations are consistent with a previous report in which MCF7 cells acquired fulvestrant resistance upon lentiviral-transduced EGFR overexpression[22]. Both MAPK and TEAD activity were inversely correlated with ERα activity across the 14 models (Supplementary Fig. 10f).

We tested whether either pathway would be targetable in the context of GiredR cell lines. We treated parental and GiredR cells with the clinically-approved MEK inhibitor, cobimetinib, and the investigational TEAD inhibitor, GNE-7883[43]. GiredR-MCF7 cells exhibited profound sensitivity to both inhibitors, with IC50 of -100 nM and 10 nM, respectively (Fig. 4i). 0810R-MCF7 clones were also sensitive to cobimetinib (Supplementary Fig. 7f). By contrast, parental MCF7 cells were insensitive to TEAD inhibition and showed some sensitivity to cobimetinib at higher concentrations (Fig. 4i). GiredR-T47D cells exhibited weaker responses. GiredR-T47D cells had increased sensitivity to cobimetinib and GNE-7883 compared to parental, but its IC50 for cobimetinib was -1 μM, and IC50 for GNE-7883 was not reached. Similarly, the cobimetinib IC50 in 0810 R cells was -5 μM for one of the two clones (Supplementary Fig. 7f). Thus, giredestrant-resistant cells have increased sensitivity to MAPK and TEAD inhibition, although the degree varies by model.

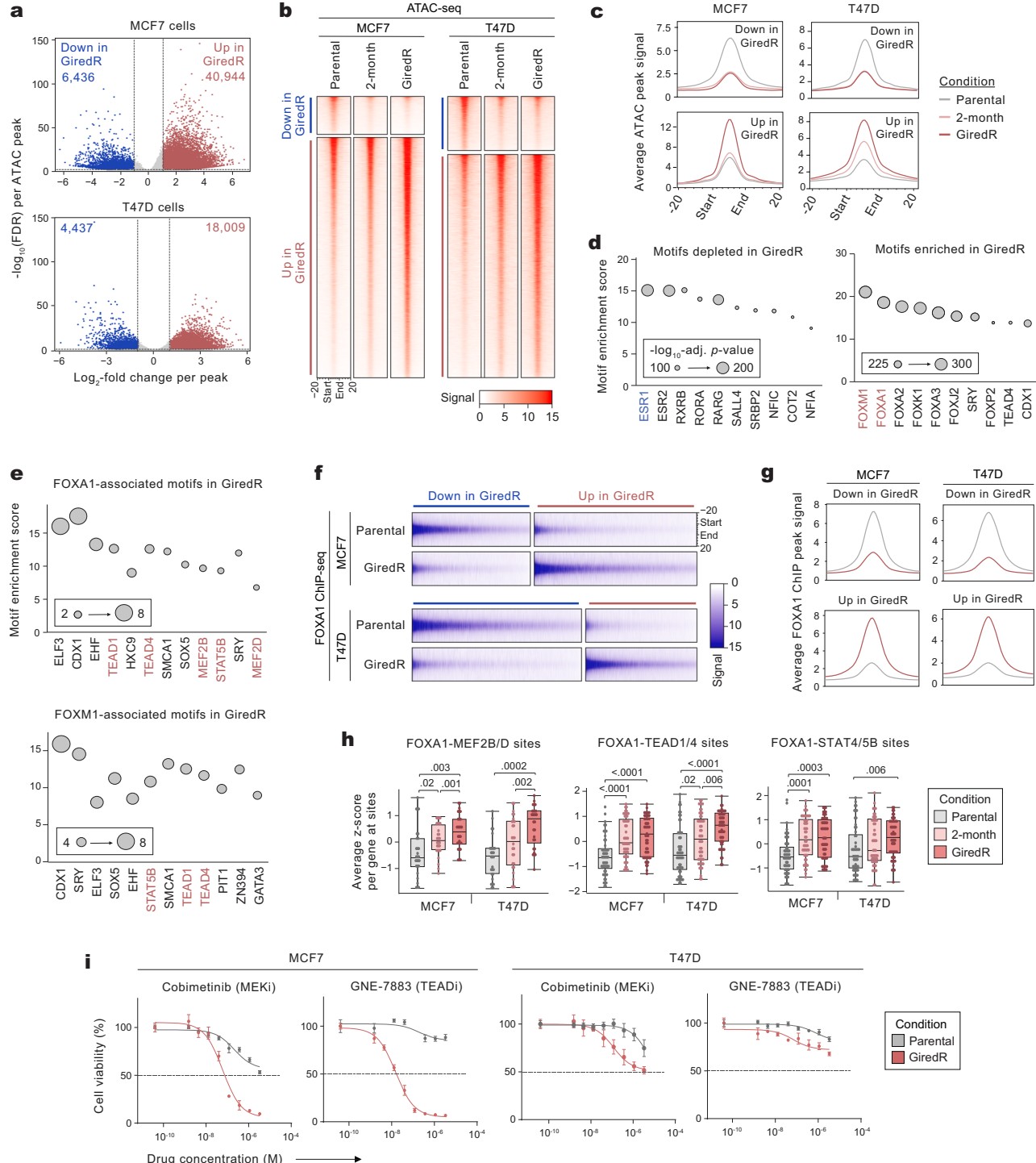

**Fig. 4 | Upon ERα inhibition, FOXA1 binds to chromatin sites associated with ERα-independence and NR pathways. a** Differential ATAC peaks (chromatin accessibility) in parental vs. GiredR cell lines for MCF7 and T47D; $n = 3$ biological replicates each. Each point is one peak; $N = 139,386$ and $128,886$ total sites, respectively. Dotted lines: $\log_2$-fold change of $\pm 1$, FDR < 0.05. **b, c** Differential ATAC peaks over treatment: parental, 2 months, and GiredR (4 months); one sample is shown, representative of $n = 3$ biological replicates. **b** Heatmaps of differential sites. **c** Average signal across differential ATAC peaks. Peaks are normalized to $\pm 20\%$ of their total length. (**d-e**) Motif enrichment analyses on differential ATAC peaks common to MCF7 and T47D (Supplementary Fig. 8a). Each point is the enrichment score for one motif. *P*-values by one-sided Fisher's exact test. **d** Motif enrichment for GiredR-depleted motifs (down in GiredR) and GiredR-enriched motifs (up in GiredR). Unchanged ATAC peaks ($n = 152,504$; defined by $-1 < \log_2$-fold change <1 and FDR < 1) were used as the reference. **e** Motif enrichment for the subset of 'up in

GiredR' peaks containing a FOXA1/FOXM1 motif. The corresponding FOXA1/M1 motif-containing 'down in GiredR' peaks were used as the reference. **f, g** Differential FOXA1 ChIP-seq peaks (Supplementary Fig. 9b) in parental vs GiredR cells; one sample is shown, representative of $n = 2$ biological replicates. **f** Heatmaps of differential peaks. **g** Average signal across differential FOXA1 ChIP-seq peaks; x-axes/legends as in (**c**). **h** For sites with co-occurrence of FOXA1 and other motifs, the expression of genes at these sites per condition. Each point is one gene; genes were included in the analysis if the gene was proximal to >1 site with the respective motif pair. $N = 46$, 52, and 52 genes for FOXA1-MEF, TEAD, and STAT sites, respectively. *P*-values by paired two-sided Mann-Whitney *U*-test. Boxes are 25th percentile, median, and 75th percentile, whiskers are maximum/minimum values within 1.5-times IQR. **i** Cell viability for parental and GiredR cells; $n = 4$ biological replicates each. Dotted lines: IC50. Values are mean ± error. Source data is available.

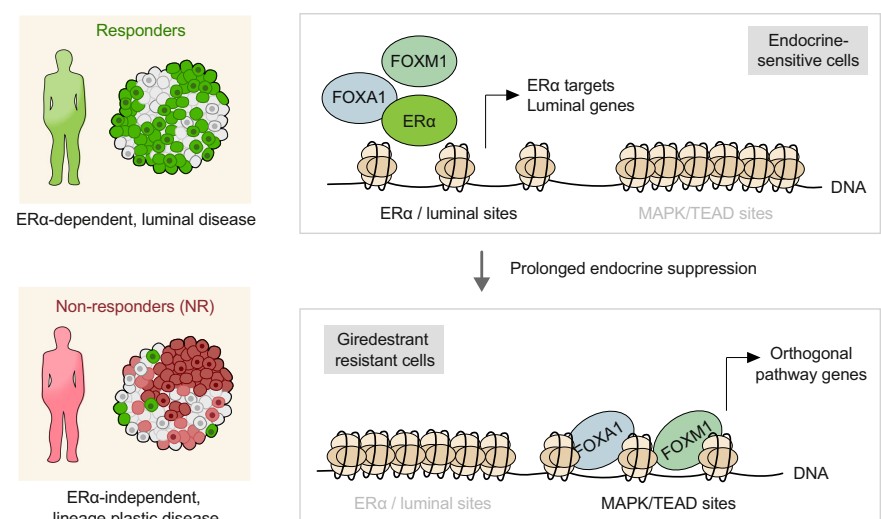

**Fig. 5 | The shifting epigenetic landscape in ERα-independent, giredestrant-resistant disease.** (Left) Patients who respond to giredestrant exhibit ERα-dependent, highly luminal disease in ER⁺ mBC. By contrast, non-responding (NR) patients lose ERα dependence and luminal features, and have elevated expression of target genes in orthogonal proliferation pathways, such as MAPK and TEAD.

(Right) In vitro, the giredestrant-resistant state is marked by repression of ERα- and luminal lineage-associated chromatin regions and inhibition the corresponding transcriptional program. FOXA1/FOXM1-associated, ERα-independent chromatin regions become opened, enabling expression of orthogonal pathway target genes.

## Discussion

Additional clinical trials for next-generation oral SERDs are underway, expanding into multiple therapeutic settings (e.g. adjuvant early cancer, 1L metastatic cancer) and combinations. Success of these trials will depend on a precise understanding of SERD response to inform patient selection. In our cohort of late-line ER⁺ mBC, we find that therapeutic response occurs in tumors that retain ERα activity and luminal lineage identity (Fig. 5). Lack of a standard-of-care arm in this phase-1a/b cohort precludes an assessment of whether luminal identity is predictive of improved outcomes for giredestrant over standard-of-care therapy. Tumors resistant to giredestrant exhibited loss of ERα dependence and luminal identity, and up-regulated orthogonal pathways including MAPK and TEAD. Long-term giredestrant drives major shifts in chromatin accessibility in vitro, resulting in the loss of luminal identity. This represses ERα binding sites, and FOXA1/FOXM1 are recruited to ERα-independent sites to drive the expression of resistance-linked genes. By contrast to *ESR1* mutations (which confer resistance to aromatase inhibitors by enabling ligand-independent ERα activation), these features are likely to promote inter-class resistance to a wide range of endocrine therapies since they bypass the need for ERα signaling altogether (Fig. 3b, Supplementary Fig. 7a). The data also predict giredestrant may offer greater benefit in less-pretreated patients whose tumors maintain dependence on ERα. Supporting this, the phase-3 lidERA BC study (NCT04961996) showed improved outcomes for giredestrant over standard endocrine therapies in the adjuvant setting.

In the context of next-generation SERD treatment, *ESR1* status is not associated with response. For GO39932, in which all patients were administered giredestrant, *ESR1*-mutant and NMD cases had similar proportions of NR and Resp (Supplementary Fig. 3e). This observation is seemingly paradoxical, since *ESR1* mutational status is strongly associated with improved PFS for multiple next-generation SERDs compared to standard-of-care endocrine therapies, leading to elacestrant's approval for *ESR1*-mutated disease. However, a closer interrogation of outcomes in the recent studies (acelERA BC, EMERALD, SERENA-2, and EMBER-3) reveals that *ESR1*-mutant patients receiving standard-of-care endocrine therapies have worse median PFS compared to *ESR1*-NMD patients. Giredestrant[9,10,14], elacestrant[11], camizestrant[13] and imlunestrant[19] each had improved clinical activity

versus standard-of-care treatment in patients with *ESR1*-mutant disease. This improvement closes the PFS gap between *ESR1*-mutated and NMD cases. For example, the median PFS for *ESR1*-mutant patients in acelERA BC receiving standard-of-care was 3.8 months, *ESR1*-mutant patients receiving giredestrant was 5.3 months, and all patients (ITT; includes both *ESR1*-mutant and NMD) was 5.4 and 5.6 in respective arms[17]. Across studies and arms, between 30-50% of *ESR1*-mutant cases progressed within 2 months (NR). Similar percentages of NR were present in the respective ITT populations.

In GO39932, patients who received giredestrant plus palbociclib (previously CDK4/6i-naïve) experienced improved PFS (Supplementary Fig. 3a–c). These results parallel that of the EMBER-3 trial, which evaluated imlunestrant (a next-generation oral SERD) as both as a monotherapy and in combination with abemaciclib[19]. Whereas monotherapy responders had highly ERα-active and luminal tumors, patients who exhibited response in the palbociclib combination arm exhibited more heterogeneity in these features. We hypothesize that while ERα dependency is linked to response in the context of SERD monotherapy, the combination with a CDK4/6i provides orthogonal suppression of tumor growth, including when ERα activity is relatively low.

The emergence of resistance/NR pathways in ER⁺ mBC poses a major challenge. Our clinical and in vitro data identify resistance mechanisms that may be actionable. In particular, our in vitro experiments provide a proof-of-concept for targeting resistant disease with EGFR/MAPK- and TEAD-targeted therapies. Acquired sensitivity to MAPK and TEAD inhibition varied across the two models evaluated. Differences in genetic background between cell lines may determine which NR pathway emerges in the context of giredestrant resistance. The interplay between NR pathways also warrants further study. In addition, advances in prostate cancer suggest that targeting factors driving lineage plasticity may also represent a viable therapeutic strategy. The PRC2 complex is known to promote lineage plasticity in prostate cancer[44]. Inhibitors of the PRC2 catalytic subunit EZH2, such as mevrometostat and ORIC-944, are being tested in metastatic prostate cancer (NCT06551324, NCT05413421). Additional work is needed to identify the targetable drivers of lineage plasticity in ER⁺ BC.

Our findings also highlight the need for better patient selection approaches. Currently, patients are screened using liquid biopsy mutational panels to assess biomarkers for treatment eligibility and

predict therapeutic benefit (e.g. *ESR1*, *PIK3CA* mutations). Here, the gain of resistance was rarely explained by acquired mutations in the respective signaling pathways. Mutational screening therefore, fails to identify the majority of mBC patients who would (1) be resistant to next-generation SERDs and (2) derive benefit from alternative or combination therapies. Emerging liquid biopsy-based approaches are promising alternatives for patient screening in the future. One such method is cell-free fragmentomics[45,46], which measures tumor gene expression and epigenetics using non-invasive liquid biopsies. Use of such approaches would help physicians identify mBC patients likely to benefit from next-generation SERDs without the need for invasive tumor biopsies.

## Methods

Our research complies with all relevant ethical regulations. Study GO39932 (NCT03332797) was conducted in accordance with the Declaration of Helsinki and Council for International Organizations of Medical Sciences International Ethical Guidelines, the International Council for Harmonization Good Clinical Practice guidelines, or applicable laws and regulations of each country where the research was conducted, if they provided greater protection to the individual. Study GO39932 exclusively enrolled female patients; all patients provided written informed consent and were not compensated for their participation. The protocol, protocol amendments, informed consent form, Investigator Brochure, and other relevant documents (for example, advertisements) were submitted to an Institutional Review Board (IRB)/Independent Ethics Committee (IEC) by the investigator and reviewed and approved by the IRB/IEC before the study was initiated. The study was conducted in 23 clinical sites, and each site reviewed the protocol via its own institution's ethics committee and criteria; see Supplementary Table 1 for full list of institutions and IRBs. Any amendments to the protocol required IRB/IEC approval before implementation of changes made to the study design, except for changes necessary to eliminate an immediate hazard to study patients. Patients did not receive intervening treatment prior to collection of baseline samples. All samples were tested in accordance with patient consent.

### Histology and immunohistochemistry

H&E and immunohistochemistry were completed by HistoGeneX (now CellCarta, www.cellcarta.com). Estrogen receptor: SP1 on Ventana Benchmark ULTRA according to package insert. Progesterone receptor: 1E2 on Ventana Benchmark ULTRA according to package insert. KI67: VENTANA CONFIRM kit (30-9) on Ventana Benchmark ULTRA according to package insert. Definiens digital image analysis was scored as a continuous percentage of positively-stained tumor cells.

### H-score calculation for tumor immunohistochemistry

Immunohistochemistry-labeled slides are reviewed by a trained pathologist. For the marker of interest, staining intensity is categorized into four possible buckets: 0 (negative), 1+ (weak), 2+ (intermediate), and 3+ (strong). H-score is calculated using the following formula: H-score = 1 x (% of intensity 1+ cells) + 2 x (% of 2+ cells) + 3 x (% of 3+ cells). A maximum score of 300 can be achieved, which reflects 100% of the tissue section having 3+ intensity for the marker of interest.

### RNA sequencing

For FFPE tumor biopsies, the H&E slide stained at HistoGeneX (now CellCarta) was used as a guiding slide for microdissection. After microdissection of the tumor area, RNA was extracted from 4 to 5 slides of 5 microns. An elution volume of 50 μL was used, and total RNA was quantified with RiboGreen. RNA-seq was performed by Q2 Solutions - EA Genomics using the TruSeq RNA Access method (www. q2labsolutions.com/en/genomics-laboratories). Total RNA library preparation and coding RNA library enrichment were followed by paired-end sequencing (2 × 50b, 50 M reads) using the Illumina sequencing-by-synthesis platform.

For cell line experiments, total RNA was extracted from cells using Qiagen's AllPrep kit as per manufacturer's protocol. Quality control of samples was done to determine RNA quantity and quality prior to their processing by RNA-seq. The concentration and the integrity of total RNA samples were determined using a 2200 TapeStation instrument (Agilent Technologies). 1 μg of total RNA was used as an input material for library preparation using TruSeq RNA Sample Preparation Kit v2 (Illumina). Library size was confirmed using the 2200 TapeStation and High Sensitivity D1K screen tape (Agilent Technologies), and their concentration was determined by qPCR based method using Library quantification kit (KAPA). The libraries were multiplexed and then sequenced on HiSeq2500 (Illumina) to generate 30 M of single end 50 base pair reads.

### Digital droplet PCR

Digital droplet PCR (Inostic BEAMing) was performed by Sysmex Inostics; see https://sysmex-inostics.com for full methods. Cell-free DNA was isolated from liquid biopsies using the QIAamp DSP Circulating Nucleic Acid purification kit (Qiagen). Genomic DNA from plasma samples was quantified by LINE-1 quantitative real-time PCR and used as a template for pre-amplification under high-fidelity PCR conditions. The pre-amplified DNA is used for the subsequent BEAMing assay. Pre-amplified DNA fragments are amplified by emulsion PCR on the surface of magnetic beads in water in oil-emulsions, then hybridized to fluorescent probes and quantified by flow cytometry. The result is the fraction of mutant DNA alleles to wild-type DNA alleles present in a particular sample, calculated by dividing the amount of mutant beads by the total amount of beads with PCR product. *ESR1*, *PIK3CA*, and *AKT1* mutations were evaluated for each sample. Specific variants screened: *ESR1*−E380Q, S463P, V534E, P535H, L536H, L536P, L536R, L536Q, Y537N, Y537S, Y537C, and D538G; *PIK3CA*−C420R, E542K, E545K, E545G, Q546K, M1043I, H1047Y, H1047R, and H1047L; and *AKT1*−E17K.

### Genomic profiling by liquid biopsy

Genomic profiling (GP) of plasma samples was performed with the FoundationACT assay in a Clinical Laboratory Improvement Amendments (CLIA)-certified, College of American Pathologists (CAP)-accredited reference laboratory (Foundation Medicine, www. foundationmedicine.com). FoundationACT uses hybrid-capture, adapter ligation-based libraries to identify genomic alterations (base substitutions, small insertions and deletions, copy number alterations, and select rearrangements/fusion events) in 66 cancer-related genes[47]. Of the plasma samples tested, 85 samples collected at baseline and 38 samples collected at EOT were successfully sequenced, having passed quality control parameters for DNA concentrations, library preparation, tumor purity, and target region coverage. Of these, 28 cases were paired, representing patients with successful sequencing of both baseline and EOT plasma samples.

### Cell culture

MCF7 and T47D cells cultured at 37 °C and 5% $CO_2$, in RPMI-1640 media supplemented with 10% FBS and 2 mM L-glutamine. Cell line authentication was conducted by Genentech's centralized cell repository. Short tandem repeat (STR) profiles were determined for each line using the Promega PowerPlex 16 System. This is performed once and compared to external STR profiles of cell lines (when available) to determine cell line ancestry. SNP profiles were performed each time new stocks were expanded for cryopreservation. Cell line identity was verified by high-throughput SNP profiling using Fluidigm multiplexed assays. SNPs were selected based on minor allele frequency and presence on commercial genotyping platforms. SNP profiles were compared with SNP calls from available internal and external data (when available) to determine or confirm ancestry.

## Generation of drug-resistant cell line models

**GiredR.** To establish GiredR cells, parental MCF7 and T47D cells were treated with 100 nM giredestrant (prepared in DMSO) for 4 months. GiredR cells were maintained in culturing media containing 100 nM giredestrant. Prior to experiments, giredestrant is removed from culturing media for 48 h.

**0810 R.** To establish 0810 R cells, parental MCF7 and T47D cells were treated with 0.1 μM GDC-0810 (prepared in DMSO). The concentration of GDC-0810 was increased stepwise over the course of 6 months to a final concentration of 8 μM. Individual clones were subsequently isolated by serial dilution or FACS. 0810 R clones were maintained in culturing media containing 8 μM GDC-0810.

## Cell viability

Cell viability is evaluated using the CellTiterGlo assay (Promega, G7570), performed per manufacturer's protocol using an EnVision plate reader (PerkinElmer, 2104-0010 A). Resistant clones are re-plated and maintained in the absence of drug for 24 h. Then, the respective drugs are added to culture at specified concentrations, and viability measurement is taken after 7 days.

## Protein analysis for cell lines

Protein extracts were prepared with RIPA Lysis Buffer (EMD Millipore). For immunoblotting, resistant cells are re-plated, maintained in the absence of giredestrant/GDC-0810 for 24 h, treated for 24 h, and then harvested. The following antibodies were used: ERα (ThermoFisher, clone SP1, MA5-14501), FOXM1 (Cell Signaling, clone D3F2B, 20459), and β-actin (Cell Signaling, clone 8H10D10, 3700). Reverse phase protein array (RPPA) was conducted by Theranostics Health for MCF7 parental and SR clones 1 and 2. Protein lysates from each cell line were diluted to a final concentration of 0.5 mg/mL, and samples were printed in duplicate on slides. The intensity value for each end point was determined within the linear dynamic range of staining after background subtraction (local background and slide staining with secondary antibody only). Intensity values were used to compute $\log_2$-fold change values per protein species across experimental conditions.

## FOXM1 siRNA

Per reaction, approximately 2 million cells were transfected with 30 pmol of either non-targeting control or FOXM1 siRNA. Cells were transfected using the Amaxa cell line nucleofector kit V (Lonza, VCA-1003) and electroporated using Nucleofector program P-020 per manufacturer's protocol. Cells were seeded at a density of 0.1 million/well in a 6-well plate format, then allowed to recover for 24 h. Following transfection, cells were grown in an Incucyte instrument to enable monitoring of cell growth rates via phase contrast, with imaging every 4 h using a 10X objective. Cells were collected at the end of the 6-day monitoring period to confirm FOXM1 knockdown via Western blot analysis.

siRNA oligonucleotides were synthesized by Horizon Biosciences; control siRNA was a pool of four non-targeting guides: UGGUUUA CAUGUCGACUAA, UGGUUUACAUGUUGUGUGA, UGGUUUACAUGU UUUCUGA, and UGGUUUACAUGUUUUCCUA; FOXM1 siRNA: CCAA CAAUGCUAAUAUUCA, CAUUGGACCAGGUGUUUAA, GCGCACGGCG GAAGAUGAA, and UGAAAGACAUCUAUACGUG.

## Whole exome sequencing

Genomic DNA was quantified with Qubit dsDNA HS Assay Kit (ThermoFisher, Q32851) and quality was assessed using Genomic DNA ScreenTape on 4200 TapeStation (Agilent, 5067-5582). For sequencing library generation, the SureSelectXT HS2 DNA System (Agilent) was used with an input of 200 ng of genomic DNA. Libraries were quantified with Qubit dsDNA HS Assay Kit, and the average library size was determined using D1000 ScreenTape on 4200 TapeStation. 750 ng of

the library is hybridized to capture probes from the SureSelect Human All Exon V7 kit (Agilent, 5191-4004). Captured libraries were quantified with Qubit dsDNA HS Assay Kit, and the average library size was determined using D1000 ScreenTape on 4200 TapeStation. Captured libraries were pooled and sequenced on NovaSeq 6000 (Illumina) to generate 50 millions paired end 75-base pair reads for each sample.

## ATAC sequencing

Nuclei were isolated from ~25,000 cells per population, sorted by FACS. For each sample, the transposition reaction was performed in a 50 μL volume using 2.5 μL Nextera Tagment DNA enzyme (Illumina, 15027865) and 25 μL of Nextera Tagment DNA buffer (Illumina, 15027865). Reaction was incubated in a 37 °C thermomixer at 1000 rpm for 30 min, then terminated by adding 5 μL of stop buffer (5 M NaCl, 1 M EDTA). DNA was purified and eluted in 15 μL volume using the MinElute Reaction Cleanup kit (Qiagen, 28204). Transposed DNA was stored at −80 °C for up to two weeks prior to library preparation.

Partially amplified libraries were generated by amplifying the purified transposed DNA for 5 cycles using NEBNext High-Fidelity 2X PCR Master Mix (New England Biolabs, M0541L). Partially amplified libraries were quantified by qPCR to determine the cycle number at which each sample produced 25% of maximum fluorescent intensity. The fluorescent intensity was used to estimate the number of additional PCR cycles required for each sample. Fully amplified libraries were purified using the SPRIselect beads (Beckman Coulter, B23317). Purified final libraries were quantified using the Qubit dsDNA HS Assay Kit (ThermoFisher, Q32851) and profiled using the Bioanalyzer High Sensitivity DNA Kit (Agilent Technologies). Libraries were pooled and sequenced on a HiSeq 2500 or NovaSeq 6000 instrument (Illumina) to generate 30 million paired-end 50-base pair reads per library.

## Chromatin immunoprecipitation (ChIP) sequencing

Per biological replicate, approximately 3 million cells were used as input for ChIP-seq. Cells were double crosslinked with 50 mM disuccinimidyl glutarate (ProteoChem, C1104) for 30 min, followed by 10 min of 1% formaldehyde. Formaldehyde was quenched by the addition of glycine. Nuclei were isolated with ChIP lysis buffer (1% Triton x-100, 0.1% SDS, 150 mM NaCl, 1 mM EDTA, and 20 mM Tris, pH 8.0). Nuclei were sheared with a Covaris sonicator using the following setup: fill level: 10, duty cycle: 15, PIP: 350, cycles/burst: 200, and time: 8 min. Sheared chromatin was immunoprecipitated overnight with the respective antibodies. Antibody-chromatin complexes were pulled down with Protein A magnetic beads and washed twice in IP wash buffer I (1% Triton, 0.1% SDS, 150 mM NaCl, 1 mM EDTA, 20 mM Tris, pH 8.0, and 0.1% NaDOC), twice in IP wash buffer II. (1% Triton, 0.1% SDS, 500 mM NaCl, 1 mM EDTA, 20 mM Tris, pH 8.0, and 0.1% NaDOC), twice in IP wash buffer III (0.25 M LiCl, 0.5% NP-40, 1 mM EDTA, 20 mM Tris, pH 8.0, 0.5% NaDOC), and once in TE buffer (10 mM EDTA and 200 mM Tris, pH 8.0). DNA was eluted from the beads by vigorous shaking for 20 min in elution buffer (100 mM NaHCO3, 1% SDS). DNA was de-crosslinked overnight at 65 °C and purified with MinElute PCR purification kit (Qiagen, 28004). DNA was quantified by Qubit, and 2 ng of DNA was used for sequencing library construction with the Ovation Ultralow Library System V2 (Tecan, 0344NB-A01) using 15 PCR cycles according to the manufacturer's recommendations. Libraries were sequenced with Illumina NextSeq, using paired-end 50-bp read configuration. Antibody clones used for ChIP-seq: anti-FOXA1 (Abcam, clone EPR10881, ab170933) and anti-FOXM1 (GeneTex, GTX102170).

## Bioinformatics data analysis
### RNA sequencing
**Processing reads.** Raw reads were processed using the HTSeqGenie workflow. Briefly, reads with low nucleotide qualities (70% of bases

with quality <23) or matches to rRNA and adapter sequences were removed. The remaining reads were aligned to the human reference genome, GRCh38.p10, using GSNAP[48,49] (v.2013-10-10-v2), allowing maximum of two mismatches per 75 base sequence. Transcript annotation was based on the Gencode human database, GENCODE 27. To quantify gene expression levels, the number of reads mapping unambiguously to the exons of each gene was calculated.

**Analysis.** Limma-voom in R was used for differential gene expression analysis. For patient data, z-scores per patient were computed from voom-normalized gene expression values and further normalized to a reference cohort of 138 ER⁺ breast cancer cases. Cell line data were not normalized to the reference breast cancer cohort. For each pairwise comparison between conditions, log₂ fold change per gene was calculated. Gene set enrichment analysis was performed using the 'fgsea' method with the Broad Institute's MSigDB gene signatures[50–52]. Heatmaps were plotted with the ComplexHeatmap package. Transcription factor enrichment analysis was performed using the ChEA3 platform[30].

**Mutation inference.** Tumor mutational status was inferred from each RNA-seq sample by detecting the presence of mutant alleles in the respective BAM files. Samples were categorized as mutant if at least 5% of the corresponding nucleotides encoded the mutant allele. *ESR1* mutational status was assessed using RNA-seq reads of the *ESR1* gene, for the following clinical variants: *E380Q, S463P, V534E, P535H, L536H/P/R/Q, Y537N/S/C*, and *D538G*. *PIK3CA* mutational status was assessed using RNA-seq reads of the *PIK3CA* gene, for the following clinical variants: *C420R, E542K, E545K/G, Q546K, M1043I*, and *H1047Y/R/L*.

**Gene signature scores.** Gene signature scores were computed as the average of z-scores for all genes defined in each signature. Gene signatures were defined in previous studies—ERα-induced genes[6]: *OLFM1, NXPH3, AMZ1, CELSR2, CT62, RBM24, FKBP4, SGK3, PPM1J, FMN1, IGFBP4, AREG, RAPGEFL1, PGR, RET, TFF1, ZNF703, RERG, SLC9A3R1, GREB1*, and *NOS1AP*; ERα-repressed genes[6]: *STON1, EGLN3, FAM171B, LIPH, SSPO, BAMBI, NBEA, GRM4, PNPLA7, DDIT4, TP53INP2, TGFB3, PSCA, BCAS1, CCNG2, TP53INP1*, and *SEMA3E*; luminal TF genes[26]: *XBP1, ESR1, FOXA1, ZNF552, PGR, RARA, GATA3, CXXC5, AR, DACH1, ZNF703, ZNF467, SPDEF, ZBTB42*, and *CREB3L4*; MAPK target genes[53]: *SPRY2, SPRY4, ETV4, ETV5, DUSP4, DUSP6, CCND1, EPHA2*, and *EPHA4*.

**Genomic profiling**
**Processing reads.** Processed mutational calls generated by FoundationMedicine; read processing, variant calling, and filtering, and copy number alteration calling were performed as previously described[47]. Per sample, each gene in the assay panel was categorized as either non-mutant or mutant. Specific alleles and alteration type (e.g short variant, amplification/deletion) were provided for mutation calls.

**Whole exome sequencing**
**Processing reads.** Raw reads were processed using the GATK4 Best Practices workflow for whole exome somatic variant discovery and annotation, on human genome (GRCh38). BAM files resulting from the pipeline were used to determine copy number variation for GiredR samples using respective parental sample as the reference. Variant calling was performed on GiredR cells relative to a parental cell reference sample, using Mutect2 from GATK (v4.1.4.1). Mutational calls were processed per sample using base R v4.1.0 tools. Copy number variation calls were assessed using the CNVkit pipeline[54].

**Analysis.** Mutations were considered cancer-associated if the corresponding gene is included in the OncoKB cancer gene list[55]. For copy number variation, a gene was considered amplified if the log₂-copy number ratio between GiredR and parental was ≥ 1 and loss if the ratio was ≤ −1.

**ATAC- and ChIP sequencing**
**Processing reads.** Raw reads were processed through the ENCODE ATAC-seq or ENCODE ChIP-seq pipelines, respectively, using MACS2 criteria for peak calling[56,57] (statistical cutoff: $p$-value $> 1 \times 10^{-6}$). Reads were mapped to human genome GRCh38.

**Analysis.** To perform differential peak analysis, the DiffBind package was used to calculate log₂-fold change and FDR values for each peak between two conditions. Heatmaps and average peak profiles were computed and plotted using the profileplyr package. For motif analyses, the MEME suite[58] (5.3.3) tools were used—(1) 'analysis of motif enrichment' (AME) with the human HOCOMOCO v11 FULL motif database[59] to identify motifs within peaks of interest and (2) 'find individual motif occurrences' (FIMO) to map motifs to individual peaks. Due to input size limits, a maximum of 60,000 peaks were evaluated in each AME/FIMO analysis; if a set of peaks exceeded this limit, 60,000 peaks were randomly-sampled to use as input for analysis. AME enrichment scores ('% enrichment over reference') were calculated from AME output as: %TP (true positive) - %FP (false positive). To map peaks to genomic features and genes associated with peaks, the chipenrich package was used.

**Statistical analysis**
Statistical analyses of experimental data were performed in R (v. 4.3.1) or Graphpad Prism 10. *P*-values are reported in respective figure panels, and statistical tests are reported in respective figure captions.

**Reporting summary**
Further information on research design is available in the Nature Portfolio Reporting Summary linked to this article.

## Data availability

Raw and processed files from sequencing experiments have been uploaded to the NCBI Gene Expression Omnibus under the following accession IDs: GSE295127 (RNA-seq; www.ncbi.nlm.nih.gov/geo/query/acc.cgi?acc=GSE295127), GSE295128 (ATAC-seq; www.ncbi.nlm.nih.gov/geo/query/acc.cgi?acc=GSE295128), and GSE305432 (ChIP-seq; www.ncbi.nlm.nih.gov/geo/query/acc.cgi?acc=GSE305432). Due to data privacy laws and the terms of informed consent for study GO39932, raw data from patient-derived F-ACT and RNA-seq cannot be deposited in a public repository nor made available upon request. The remaining data for Figs. 1–4 and all Supplementary Figs. have been provided as Source Data files. Source data are provided with this paper.

## Code availability

Analysis code is available at https://doi.org/10.5281/zenodo.18644938. For ATAC-seq and ChIP-seq, all analysis code uses publicly available R packages. For RNA-seq, some analyses code includes custom in-house functions. The custom code was designed to analyze RNA-seq data within our institution's cloud-based computing environment, and will not function properly outside of this environment. However, comparable analyses can be performed using publicly available R packages.

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

## Acknowledgements

We would like to thank Ann E. Collier, Alejandro Chibly, and Xiaosai Yao for helpful discussions; Meritxell Bellet, Sherene Loi, Aditya Bardia, Valentina Boni, Joohyuk Sohn, Tomas G. Neilan, Rafael Villanueva-Vázquez, Peter Kabos, Laura García-Estévez, Elena López-Miranda, Jose A. Pérez-Fidalgo, and Jose M. Pérez-García for providing biopsies in the clinical trial (NCT03332797); and patients for their participation in the clinical trial. This study was supported by funding from Roche/Genentech.

## Author contributions

J.L. designed/conducted the biomarker analyses and wrote the manuscript. J.L. & C.M. designed the experiments. K.H., C.O., & J.Guan performed the experiments. V.K. and B.D. performed the ChIP sequencing, M.S. and Y.L. performed the RNA and ATAC sequencing, J.Giltnane performed pathology review. H.M.M., J.A., C.-W.C., M.R.G., J.E.-W., P.P.-M., K.L.J, N.C.T, & E.L. conducted the clinical study. All authors reviewed the manuscript.

## Competing interests

The authors declare the following competing interests: excepting K.L. Jhaveri, N.C. Turner, and E. Lim, all authors are Genentech/Roche employees and own shares of Roche. C.M. is a named co-inventor on patent 11081236 entitled "Diagnostic and therapeutic methods for the treatment of breast cancer". K.L. Jhaveri has a consultant/advisory board role in Novartis, Pfizer, BMS, Jounce Therapeutics, Taiho Oncology, Genentech, Inc., AbbVie, Eisai, AstraZeneca, Blueprint Medicine, Daiichi Sankyo, Sun Pharma Advanced Research Company Ltd, Gilead, Seattle Genetics, Olema Pharmaceuticals, Menarini/Stemline, and Lilly/Loxo Oncology; and has received research funding from Novartis, Clovis Oncology, Genentech, Inc., AstraZeneca, ADC Therapeutics, Novita Pharmaceuticals, Debio Pharmaceuticals, Pfizer, Lilly Pharmaceuticals/Loxo Oncology, Zymeworks, Gilead, Puma Biotechnology, Context Therapeutics, and Merck Pharmaceuticals. N.C. Turner has received advisory board honoraria from AstraZeneca, Lilly, Pfizer, Roche–Genentech, Novartis, GlaxoSmithKline, Repare Therapeutics, Relay Therapeutics, Zentalis, Gilead, Inivata, Guardant, and Exact Sciences; and research funding from AstraZeneca, Pfizer, Roche–Genentech, Merck Sharpe & Dohme, Guardant Health, Invitae, Inivata, Personalis, and Natera. E. Lim reports research support from Ellipses, Novartis, and Pfizer; speaker compensation from AstraZeneca, Gilead, Novartis, Lilly, Pfizer and Roche; meeting and/or travel support from AstraZeneca, Novartis, and Gilead; advisory board compensation from AstraZeneca, Gilead, Lilly, MSD, Novartis, Pfizer and Roche; and leadership or fiduciary roles in the Breast Cancer Trials and University of New South Wales.

## Additional information

[1]Department of Translational Medicine, Genentech, South San Francisco, CA, USA. [2]Department of Discovery Oncology, Genentech, South San Francisco, CA, USA. [3]Department of Proteomic and Genomic Technologies, Genentech, South San Francisco, CA, USA. [4]Department of Research Pathology, Genentech, South San Francisco, CA, USA. [5]Department of Biostatistics, Genentech, South San Francisco, CA, USA. [6]Department of Early Clinical Development, Genentech, South San Francisco, CA, USA. [7]Department of Product Development, Genentech, South San Francisco, CA, USA. [8]Breast Medicine Service, Department of Medicine, Memorial Sloan Kettering Cancer Center, Weill Cornell Medical College, New York, NY, USA. [9]Royal Marsden Hospital and Institute of Cancer Research, London, UK. [10]St Vincents Hospital, University of New South Wales and Garvan Institute, Sydney, NSW, Australia.
✉e-mail: liang.jackson@gene.com

