## [Transparent Peer Review file · Nature Communications]

Loss of luminal lineage drives resistance to next-generation ER α antagonists in pretreated ER+ HER2- locally-advanced or metastatic breast cancer

Corresponding Author: Dr Jackson Liang

Version 0:

Reviewer comments:

Reviewer #1

(Remarks to the Author)

I find this manuscript compelling for understanding the response and resistance mechanisms associated with SERDs, SERMs, CDK4/6 inhibitors, and the combination of SERD and CDK4/6 inhibitors. The laboratory work and analysis of clinical samples complement each other, creating a strong synergy in the research.

All of the reviewer's comments have been addressed accurately and systematically. Many new experiments and analyses have been incorporated, leading to valuable research findings that have been effectively presented.

In Figure 3, the response to Tamoxifen (TAM) appears bell-shaped. From a target concentration perspective, it seems that certain regulatory mechanisms may be involved.

Reviewer #4

(Remarks to the Author)

Through the revisions, Liang et al appropriately reframed analyses to account for the difference in prior CDK4/6 inhibition, more clearly demonstrating the differences in responders versus non-responders to giredestrant. Their expanded in vitro analysis across resistant cell lines adequately showed common pattern of loss of ER signaling and an up-regulation of alternative pathways, EGFR/MAPK and Hippo/TEAD. The additional ATAC-seq data exploring a mechanistic underpinnings for endocrine resistance and loss of luminal features. Demonstrating the limitations of mutation calls over ddPCR are clinically important and should be highlighted. Further delineating those patients who had prior CDK4/6i treatment vs those who are treatment naive will continue to improve the applicability of the study population to real-world patients, most of whom will have received CDK4/6 therapy. I have a few additional minor comments below.

Minor comments:

Line 201, pg6: while definitions of luminal-high and luminal-low is clearly written in the methods, it would benefit the reader to add a brief definition within the main body of the text rather than the figure legend.

Figure 4e: I could not follow what the two different barplots refer to. What is the difference between the left and right parts of 4e? They look identical with no indicating title or description in the figure legend.

Extended Data Figure 2: is it possible to add prior CDK4/6 treatment as a color bar under response? This could aid in visualization and interpretation of the figure.

Extended Data Figure 3b: for the panel demonstrating Treatment arm, is it possible to also demonstrate the survival curve within the Combo group of those who had received prior CDK4/6 inhibition?

Reviewer #5

(Remarks to the Author)

In the manuscript “Loss of luminal lineage drives resistance to next-generation ERa-antagonists in pretreated ER+ HER2-locally-advanced or metastatic breast cancer”, Liang et al. investigate the mechanistic basis for response to treatment in the phase 1a/b study with the oral selective estrogen receptor degrader (SERD) giredestrant (NCT03332797) alone or in combination with a CDK4/6 inhibitor. The authors have performed exploratory analyses of both tumor biopsies and ctDNA from patients on the trial and in addition have developed in vitro cell line models of giredestrant resistance. In the trial patients with metastatic ER+/HER2- breast cancer were treated with giredestrant at several different dose levels and a subset were treated with the combination of giredestrant plus palbociclib. Given the nature of the trial and the lack of a control arm, the results are largely hypothesis generating. Nonetheless the data from the trial are valuable and should be reported.

Major critiques:

- 1) While the primary data from the in vitro models has been deposited in GEO, the RNA-seq and liquid biopsy data from the trial should also be made available.
- 2) The authors argue that in the giredestrant resistant models that the inability to identify a clear genetic mechanism of resistance suggests that resistance is primarily epigenetically driven. The data presented do not support this conclusion. The stable resistance phenotype of GirdR models may indicate genetic selection and there may be different genetic mechanisms in MCF7 or T47D cells. In addition, there may be multiple different genetic mechanisms in distinct clones present in the GirdR cultures that may mask distinct genetic resistance mechanisms. Single cell cloning or sequencing would be needed to address this possibility. In Figure 3e, using whole exome sequencing (WES), over 50 mutations are enriched in GirdR cells, including known cancer-associated variants, but follow-up experiments are not performed. Additionally, copy number variation (CNV) analysis was performed on GirdR cells, which identified a loss on chromosome X including 19 cancer associated genes (Line 335), but this data is not presented in the figure and there is no validation. Deletions found in the CNV analysis further supports genetic selection of a SERD resistant clone. The authors state “Since few genetic mutations were acquired by GiredR cells, we considered whether epigenetic alterations may drive ER α independence and loss of luminal identity” (Line 362), and in Figure 5 they propose an epigenetic model of endocrine resistance. The authors should point out more clearly that genetic mechanisms of resistance remain a likely explanation despite their inability to identify them.
- 3) This manuscript relies heavily on computational imputation without validation. For example, in Figure 4d-f, ATACseq identified chromatin opening at FOXM1 motifs, which the authors argue is a novel driver of endocrine resistance (Line 34), however the ATACseq results do not necessarily demonstrate a functional role for FOXM1. For example, the FOXM1 motif highly resembles the FOXA1 motif, a known driver of endocrine resistance. A direct role for FOXM1 should be validated using FOXM1 ChIPseq and testing whether FOXM1 is necessary for resistance in GirdR cells and sufficient to induce resistance in parental cells.
- 4) The use of the FOXM1 inhibitor FDI-6 is not convincing. What is the evidence that its effects are on-target? Can the effects be reproduced genetically?
- 5) The model presented in Figure 5 is not supported by the data presented.

Other questions and minor concerns:

- 1) Are any ESR1 gene amplifications, deletions, or gene fusions identified in patients?
- 2) The methods section indicates that WES was performed using a mouse genome kit (Line 1045).
- 3) Please define NMD (Line 128) and specify the genes for the Luminal TF (Figure 1F) and the MAPK signature (Ext Data Fig 9a) in this manuscript, and update Ref #26.

Version 1:

Reviewer comments:

Reviewer #4

(Remarks to the Author)

I have no further comments for the authors. The prior comments raised have all been adequately addressed, and the manuscript should be shared with the broader community.

Reviewer #6

(Remarks to the Author)

This study addresses an important clinical issue: the variable response to next-generation SERDs in advanced ER+/HER2- breast cancer. Using multi-omics data from a giredestrant clinical trial and in vitro resistance models, the authors conclude that loss of luminal lineage identity, rather than ESR1 mutations, underlies resistance to ER α -targeted therapy. The study is strong, methodologically rigorous, and offers meaningful mechanistic and clinical insights. However, some key concerns are needed to be addressed.

Major Comments:

1. The authors provided strong evidence for the role of FOXA1 in driving a resistant phenotype through chromatin remodeling. However, the mechanism by which FOXA1 expression and activity are upregulated in the resistant cells

remains unclear. Is this a stochastic event, or is it driven by a specific upstream signaling pathway? Furthermore, the authors show that ~70% of FOXA1 binding sites in resistant cells are not co-occupied by ER α . What are the transcriptional partners of FOXA1 at these sites, and what are the key target genes that drive the resistant phenotype? Have the authors considered performing ChIP-seq for FOXA1 in their resistant cell lines to identify its direct targets?

2. The study focuses on a specific mechanism of resistance involving the loss of luminal lineage. However, it is likely that other mechanisms of resistance also exist. The authors themselves show that a subset of non-responders still retain some ER α activity. How do the authors envision the interplay between the mechanism they have identified and other known resistance mechanisms, such as the acquisition of ESR1 mutations or the activation of other receptor tyrosine kinases? A more nuanced discussion of the heterogeneity of resistance would strengthen the manuscript.

3. The manuscript demonstrates upregulation of multiple bypass pathways (EGFR/MAPK, Hippo/TEAD, Wnt, IL2/STAT5) in non-responder tumors. However, it remains unclear whether these pathway activations are drivers of resistance or merely consequences of losing ER α -dependent growth control. Have the authors performed functional validation experiments (e.g., genetic or pharmacological inhibition of these pathways) to determine whether blocking these pathways can restore giredestrant sensitivity?

4. The observation that different resistant cell lines (MCF7-GiredR vs. T47D-GiredR) show differential activation of bypass pathways (MAPK in MCF7 but not T47D) suggests context-dependent resistance mechanisms. What determines which bypass pathway becomes activated in a given cellular context? Are there pre-existing differences in signaling pathway wiring between MCF7 and T47D cells that predispose them to different resistance mechanisms?

5. The authors identify TEAD4 and TCF4 as transcription factors associated with non-responder genes, implicating Hippo/TEAD and Wnt signaling. However, canonical activation of these pathways typically involves nuclear translocation of YAP/TAZ (for Hippo) or β -catenin (for Wnt). Have the authors examined the subcellular localization and activation status of these key signaling mediators in resistant cells? Are there genetic alterations in pathway components that would explain constitutive activation?

6. The authors also generated resistance models using GDC-0810 (a SERD-SERM hybrid) and observed similar features. Have they tested other next-generation SERDs (e.g., elacestrant, camizestrant) to determine whether this resistance mechanism is common to all SERDs or specific to certain chemical scaffolds?

Minor Comments:

1. In Figure 2, the authors show that several transcription factors are associated with non-responder genes. It would be helpful to show the expression levels of these TFs in the responder versus non-responder patient samples.

2. The authors mention that the MCF7-GiredR cells have deletions on chromosome X. Are any of the deleted genes known to be involved in epigenetic regulation or ER signaling? A more detailed discussion of the potential functional consequences of these deletions would be welcome.

3. The sample sizes for some comparisons are relatively small (e.g., n=10 NR vs. n=17 Resp pairs for the on-treatment analysis). Have the authors performed power calculations to determine whether they have sufficient statistical power to detect meaningful differences?

4. Ensure consistent use of drug-class terminology (e.g., "next-generation ER α -antagonists," "SERD-like," "PROTAC-like"), and consistently name or classify the compound used throughout the manuscript.

Reviewer #7

(Remarks to the Author)

Reviewers' Comments:

Reviewer #1 (Remarks to the Author):

1) This study deals with an essential topic in the treatment of recurrent breast cancer, especially HR+/HER2-. It is a prospective translational study, and its significance is considered high. On the other hand, the number of cases is limited, and the number of analyses of tissue-paired samples is relatively small, so its interpretation should be done with caution.

To address concerns over small sample size, we profiled additional baseline biopsies ($n = 27$) collected on this study, which has nearly doubled our sample size from 28 to 55 biopsies. This empowered additional analyses, such as a comparison of single-agent vs. palbociclib combination-treated cases (see point #4 below), and enabled the use of other approaches to interrogate differences between responders vs. non-responders – e.g. generalized linear modeling (new Fig. 2c) and transcription factor enrichment analysis (new Fig. 2f). Statistical significance of prior analyses was also improved by these additional samples.

2) On the other hand, there are countless previous reports in this research field and a large amount of related literature and data. The consistency and inconsistency with these would be the essential evaluation points of this study.

We have now added in-line discussions of relevant literature/data in the Results section when describing our study findings.

3) The concept of treatment resistance is vague in the introduction and discussion. There is endocrine therapy (ET) resistance, in which case there is complete resistance to ET, which might be related to FP in this study), resistance about each class and intraclass resistance. This study addresses class-related resistance in interpreting the analysis results of human samples and cell line experiments. Intraclass resistance may correspond to the relationship with fulvestrant resistance and the comparison of resistance between new SERDs. It may be difficult to discuss all of these systematically because of limited data, but it would be easier to understand if you discussed them in such a concept. The analysis results seem diverse and complex, and multifactorial interpretation is required at each point. In other words, it is difficult to see the essence when considering it in an integrated manner.

In both ER⁺ tumor biopsies and *in vitro* models, giredestrant resistance was associated with loss of ER α dependence and luminal identity, and up-regulated orthogonal pathways including MAPK and TEAD. The observed features are likely to promote inter-class resistance to a wide range of endocrine therapies since they bypass the need for ER α signaling altogether (e.g. 'complete resistance to ET' as the Reviewer describes). This is by contrast to *ESR1* mutations, which primarily confer resistance to aromatase inhibitors by enabling ligand-independent ER α activity. To further explore this point, we examined cell lines which we engineered to acquire resistance to giredestrant and another investigation anti-ER α drug, GDC-0810. The two drugs represent distinct mechanisms of pharmacological ER α inhibition: whereas giredestrant is a full SERD, GDC-0810 is a SERD-SERM hybrid. The same features (loss of ER α dependence and luminal identity) emerge *in vitro* when cells acquire resistance to either drug. Both giredestrant- and GDC-0810-resistant MCF7 and T47D cells develop cross-resistance to standard-of-care ER α therapeutic ligands: the SERD fulvestrant and the SERM tamoxifen (new Figure 3, new

Extended Data Fig. 7). We have added a discussion of these points in the first paragraph of the Discussion section.

We agree that multifactorial interpretation should be considered – we address this as described in point #5 below.

4) In addition, the involvement of a CDK4/6 inhibitor must also be considered. I would like this part (involvement of CDK4/6) to be as easy to understand as possible.

This is an important point, since patients who received the giredestrant-palbociclib combination were more likely to be responders (new Extended Data Fig. 3a-c). We have now restructured our analyses to explicitly address this point. We now begin with a comparison of responders vs. non-responders in the single-agent giredestrant treatment arm (new Figure 1). Next, since the majority of patients in the palbociclib combination arm were responders (PFS > 2 months), we compared single-agent responders to palbo-combination responders (new Extended Data Fig. 4f-g, Figure 2).

While single-agent responders had highly ER α -active and luminal tumors, patients who exhibited response in the palbociclib combination arm exhibited significantly more variation in these features, comprising a mix of ER α -high and ER α -low tumors ($p = 0.001$ and 0.01 by two-sided F-test for variance). We hypothesize that while ER α dependency is linked to response in the context of giredestrant monotherapy, the combination with a CDK4/6i provides orthogonal suppression of tumor growth independent of the ER α signaling axis. We now highlight these points in paragraph 3 of the Discussion section.

5) The factors that make up the clinical effect, especially the duration of response, are multivariate. In addition to the factors discussed in this study, there are many others, such as tumour volume, heterogeneity, and the influence of previous treatment. As discussed, the immune system is also involved. It would be better to explain clearly which part was approached this time.

As the Reviewer highlights, the determinants of treatment response may be complex and multifactorial. To address this point, we performed univariate Cox proportional hazards modeling on key clinical features to assess the influence of each on progression-free survival in $n = 181$ enrolled patients (new Extended Data Fig. 3a). Features related to patient health (e.g. age, BMI, ECOG), disease characteristics (ESR1 mutational status, histological subtype, tumor stage/grade, TNM staging) and treatment history (chemotherapy, fulvestrant, CDK4/6i) were evaluated. Critically, tumor heterogeneity and the immune system were not evaluable since our clinical trial was not designed to capture these features.

The only features associated with PFS were related to CDK4/6 inhibitors (Extended Data Fig. 3a-b). Patients who received prior CDK4/6i had significantly worse PFS, and patients enrolled in the palbociclib combination arm had improved PFS. These two effects are difficult to differentiate, since most of the patients in the palbociclib combination arm had no prior CDK4/6i exposure (Extended Data Fig. 3c). These analyses, and explanation of which specific features were evaluated, are now described in paragraph 1 of the second Results section. In addition, we have now re-structured our analyses of patient data to account for the profound influence of palbociclib combination on responses (see point #4 above).

6) The study of resistant clones is intriguing. However, it is limited to one cell line, and the data is challenging to interpret, especially when considering clinical data, and the current logical development seems unclear. A more complete study could be performed if experiments were conducted using multiple cell lines and under various conditions, including *in vivo*. Still, it would not be accessible in terms of time and quantity. If your research focuses on clinical analysis, I would like you to consider how to structure the section on resistant cell lines. It would be meaningful to analyse artificially resistant cells, cloned by exposing them to high concentrations of drugs for half a year, and explore how their behaviour, characteristics, and changes due to treatment compare with clinical results, and conversely, how the dynamics within the cell line correspond to clinical results. This may be a supplemental analysis, but it will help the reader understand. Finding a treatment to overcome resistance from this research might be possible, but on the other hand, someone may feel far.

We thank the Reviewer for this suggestion. We have now greatly expanded the pre-clinical experiments on this study, including the Reviewer's suggestion of exposing cells to high concentrations of drugs and profiling their changing characteristics over the course of treatment. Deep profiling experiments were performed on two different workhorse models of ER⁺ breast cancer: MCF7 and T47D (new Figs. 3-4, Extended Data Fig. 6, 8). We additionally interrogated giredestrant resistance across a panel of 14 different ER⁺/HER2⁻ breast cancer cell lines (new Extended Data Fig. 9). The new data are limited to *in vitro* experiments, since *in vivo* studies were not feasible in terms of time or quantity as the Reviewer suggests.

The new data are described in the latter 3 sections of Results; below are some highlights:

- Over 4 months of giredestrant treatment, MCF7 and T47D cells acquired resistance to giredestrant (GiredR) via loss of ER α and luminal lineage identity, as seen in giredestrant non-responsive patients (NR). GiredR cells also acquired activity in NR-enriched pathways (e.g. MAPK, TEAD) over treatment.
- Across 14 ER⁺/HER2⁻ breast cancer cell lines, resistance to giredestrant was significantly correlated with low ER α activity and elevated MAPK/TEAD signaling.
- Assessing the chromatin landscape over treatment, we found that ER α -associated chromatin sites became suppressed, followed by the induction of ER α -independent, MAPK/TEAD sites. FOXA1 and FOXM1 (which normally regulate ER α) were significantly associated with the MAPK/TEAD sites induced in GiredR.
- GiredR cells exhibited increased sensitivity to inhibitors targeting the NR-enriched pathways, MAPK and TEAD, although the degree of sensitivity varied between models. Inhibition of FOXM1, which was implicated in both NR biopsies and GiredR cells, drove a complete loss of cell viability in both parental and GiredR cells.

Minor:

'Surprisingly, giredestrant did not significantly alter gene expression in FP cases. By contrast, giredestrant repressed a program of 135 genes in LTB tumors.' This result may not be surprising based on the previous data.

We have removed the term 'surprisingly' in the relevant text.

Reviewer #2 (Remarks to the Author):

The authors are addressing very important and timely question in a well defined cohort of patients included in a clinical trial. The data is biologically sound and novel, and the methodology used to address the questions appropriate. The statistics used is appropriate.

The flow of the work is sound and conclusions robust and clinically useful. The paper addresses the most urgent question on how to potentially segregate patients in those who benefit from endocrine treatment (SERDs) in later lines and patients who don't.

We thank the Reviewer for this assessment of our manuscript.

The only potential lack of the study is that the model used encompassed only one cell line, however results were conclusive and the main focus on the clinical analyses and it even is not really needed to improve the manuscript with additional analyses. Maybe add a little emphasis on that in the discussion.

The limited exploration in cell line models was highlighted by multiple Reviewers. We have now greatly expanded our cell line experiments to model giredestrant resistance with a molecular deep-dive on the two most common ER+ breast cancer cell lines, MCF7 and T47D. We additionally explore the association between giredestrant resistance and clinically-identified transcriptional features in a panel of 14 ER+ breast cancer cell lines. In particular, we have identified the transcriptional and epigenetic events that drive endocrine resistance and identified additional therapeutic options in pre-clinical experiments on cells with acquired resistance to giredestrant. Please see our response to Reviewer 1, comment #6 above; the new data are described in Figures 3-4 and Extended Data Figs. 6-8.

Reviewer #3 (Remarks to the Author):

In this paper, Liang et al investigated heterogenous response to next-generation SERD giredestrant +/- palbociclib (CDK4/6i) in a Phase 1a/b trial enrolling ER+HER2- metastatic breast cancer in the second or third line of therapy, identifying gene expression resistance pathways that emerged in fast progressors compared to long-term responders. Aspects of their analyses confirm expected results – comparable to those seen for example in the FELINE trial (Griffiths Nature Cancer 2021) – that fast progressors (intrinsically resistant tumors) have lower estrogen receptor pathway activity at baseline, less of a reduction of proliferation and ESR1 signaling with SERD treatment, and upregulation of orthogonal pathways including MAPK/ERK. Most results are derived from 29 patients with paired pre- and on-treatment tumor biopsies; they also draw from a larger cohort of 172 patients with some type of tumor or blood data. Overall the work is interesting, especially the suggestion that EGFR or MEK or ERK inhibition could overcome resistance in a subset of patients.

We thank the Reviewer for this assessment of our manuscript, and strongly agree with the need to identify therapeutic strategies to overcome resistance to endocrine therapies. In this revised manuscript, we have elaborated on this point by further exploring the transcriptional and epigenetic events that drive endocrine resistance and identifying additional therapeutic options

(e.g. TEAD inhibition) in pre-clinical experiments on cells with acquired resistance to giredestrant.

Major comments:

1. It's not readily apparent what the value is of the larger cohort of 172 with some type of biopsy. I do think more clarity is needed about which ones had tumor biopsy, which ones had liquid biopsy, and what timing. But in general, they do not appear to contribute substantially to a coherent story. For example – what does an early section of the results focused on concordance between tumor RNA-seq and ddPCR and F-ACT add to the main takeaways of this paper?

Given the recent approval of elacestrant for *ESR1*-mutant disease and multiple ongoing studies using *ESR1* mutational status as an enrollment criteria (e.g. evERA [NCT05306340], pionERA [NCT06065748]), we initially felt that concordance data between multiple *ESR1* assays would be of interest to the ER⁺ breast cancer community. *PIK3CA* mutation prevalence is likewise relevant for alpelisib and recently-approved inavolisib. However, we agree with the Reviewer's suggestion that the relevant section of text was excessive and detracted from the main focus of our manuscript, which is to understand why some patients benefit from next-generation SERDs while others do not. While we still feel the community may derive value from these mutation/concordance data, we have now minimized our description of the data into two sentences of the main text and abridged the data panels into Extended Data Fig. 2. In addition, we have streamlined our text and methods by removing mentions of biopsy numbers, except in instances where this information is relevant for data interpretation.

2. Grouping tumors treated with giredestrant alone or giredestrant + palbociclib into one analytic cohort introduces bias and would be helpful if this were handled more explicitly. For example, given that the LTB were more likely to be treated with palbociclib, and that you grouped tumors exposed to single-agent giredestrant and palbociclib together, how do you ensure you aren't identifying pathways associated with palbociclib treatment? While numbers are small, presumably some differences are seen in gene expression for tumors treated with giredestrant alone vs in combination with palbociclib – i.e. presumably palbociclib induces greater proliferation change (e.g. PALLET)?

This is an important point – in our cohort, prior exposure to CDK4/6 inhibitors was poorly prognostic, while patients who received the giredestrant-palbociclib combination more likely to respond (Extended Data Fig. 3a-c). These results parallel that of the recent EMBER-3 trial on imlunestrant, in which similar patients receiving the combination of imlunestrant and abemaciclib had stronger responses than imlunestrant monotherapy (HR = 0.57, $p < 0.001$).

To address this point, we have restructured our analyses to start with a comparison of single-agent responders vs. non-responders (new Figure 1). Since the majority of patients in the palbociclib combination arm were responders (PFS > 2 months), we next compared single-agent responders to palbo-combination responders (new Extended Data Fig. 4f-g, new Figure 2). By separating the single-agent responders from palbo-combination responders, we increased our confidence that the identified pathways/features were associated with giredestrant response, rather than a palbociclib effect. While single-agent responders had highly ER α -active and luminal tumors, patients who exhibited response in the palbociclib combination arm exhibited significantly more heterogeneity in these features, comprising a mix of ER α -high

and ER α -low tumors ($p = 0.001$ and 0.01 by two-sided F-test for variance). We hypothesize that while ER α dependency is linked to response in the context of giredestrant monotherapy, the combination with a CDK4/6i provides orthogonal suppression of tumor growth independent of the ER α signaling axis. We now highlight these points in paragraph 3 of the Discussion section.

3. Is it so surprising that immune signatures were enriched in the progressors? Other work has shown that these tend to correlate well with estrogen expression (i.e. more estrogen activity is correlated with less immune infiltration), so given that progressors have less estrogen activity, perhaps it makes sense they have more immune signature enrichment. Without supporting clinical or preclinical data, I would not suggest that this would necessarily correlate to greater immunotherapy responsiveness.

This is a good point – we have removed any speculation in the text regarding the responsiveness to immunotherapy, since we lack clinical or pre-clinical evidence on this topic.

4. As above, I do find the cell line work suggestive and interesting, but would urge the authors to expand it a bit more. Ideally *in vivo* studies, or at least a couple more replicates.

We thank the Reviewer for this suggestion. We have now greatly expanded our cell line experiments to model giredestrant resistance with a molecular deep-dive on the two most common ER+ breast cancer cell lines, MCF7 and T47D. We additionally explore the association between giredestrant resistance and clinically-identified transcriptional features in a panel of 14 ER+ breast cancer cell lines. In particular, we have identified the transcriptional and epigenetic events that drive endocrine resistance and identified additional therapeutic options in pre-clinical experiments on cells with acquired resistance to giredestrant. Please see our response to Reviewer 1, comment #6 above; the new data are described in Figures 3-4 and Extended Data Figs. 6-8. The new data are limited to *in vitro* experiments, since *in vivo* studies were not feasible in terms of time or quantity.

Minor comments:

1. Intro: Need a citation for “Unfortunately, poor bioavailability limits the efficacy of fulvestrant in both ESR1-mutant tumors and the broader metastatic ER+ population.”

We have now included the appropriate citations.

2. Intro: Misspelled elacestrant (in intro and discussion)

This has been corrected.

3. Intro: You mention that 30-50% were rapid progressors on trials like aceIERA, EMERALD4, etc. What percentage of patients had sustained benefit up to 30+ months in these trials?

As our earliest study, the current phase-1 giredestrant trial has had the longest follow-up time, and some patients were tracked beyond the 30 month timepoint. However, publicly-available data on the remaining studies do not extend to 30 months of follow-up. Below, I include the

percentage of ITT patients in respective investigational arms who remained on-study for the maximum duration of follow-up. In the acELERA BC study, ~15% of patients in the giredestrant arm remained on-study for 12 months. In the EMERALD study, ~15% of patients in the elacestrant arm remained on-study for 25 months. In the SERENA-2 study, ~20% of patients in the 75 mg camizestrant arm remained on-study for 25 months. In the recent EMBER-3 study, ~10% of patients in the imlunestrant arm remained on-study for 28 months. Note that each study had different eligibility criteria, so direct comparison across studies cannot be made. We have updated the language in the Introduction section to “patients had sustained benefit up to 2+ years”.

4. Intro: would include information on trial eligibility and design early in results or in intro.

We have now included this information in the first paragraph of the Results section.

5. Results: why does it say 172 for the larger cohort in the results, but 167 in the materials and methods?

Per major comment #1 above, we appreciate the confusion caused by the various biopsy numbers. To address this, we have streamlined our text and methods by removing mentions of biopsy numbers, except in instances where this information is relevant for data interpretation.

6. Results: I'm not sure how useful the comments on PAM50 subtype switch (e.g. Fig 1d) are. Proliferation is suppressed in the presence of drug, something that is sorted out better later, but this does not mean subtype was changed.

We have removed the discussion of this point from the manuscript text, and moved the relevant figure panel to Extended Data Figure 1.

7. Results: Is there any statistically significant difference of PAM50 subtype in FP vs MP vs LTB?

To address this question, we included a new figure panel to compare baseline PAM50 subtypes across response groups (new Extended Data Fig. 1f). There is no significant difference in PAM50 subtypes, and the relative proportion of LumA to LumB is similar between response groups.

8. Results: for the 135 genes suppressed on giredestrant therapy, how were these identified exactly? Were they differentially responding genes by some criterion?

The 135 genes were identified unbiasedly via differential gene expression analysis. We compared differential gene expression between the baseline and on-treatment (C2D8) biopsies from LTB patients. The analysis was performed using the limma-voom method, with statistical cutoffs as follows: \log_2 -fold change ≥ 1 (up at C2D8) or ≤ -1 (down at C2D8), FDR < 0.1 . We have revised the main text to highlight our analytical approach.

Also note that in this revised manuscript, we have followed reviewer suggestions to streamline our analyses, and have grouped the former MP (medium progressers) and LTB (long-term

benefitters) into a single group, called responders (Resp). We now include the same analysis on the Resp group, which identifies 144 genes which are suppressed by giredestrant (new Fig. 1b-c), computed using the same method as described above.

9. Results: Could you comment more (in the Discussion e.g.) on *ESR1* mutation being enriched in the fast-progressing cases, especially given existing FDA approval for oral SERD is for *ESR1* mutated tumors only?

In our cohort, *ESR1* mutations are not enriched in fast-progressing cases*. Rather, both non-responders and responders have *ESR1* mutations in ~30-40% of cases (new Fig. 2b; non-significant by χ^2 -test). By Cox proportional hazards modeling, *ESR1* mutational status is not significantly associated with differences in progression-free survival (new Extended Data Fig. 3). As the Reviewer notes, these observations initially appear paradoxical, since the FDA approval for oral SERD is specific to *ESR1*-mutated disease. However, upon closer examination of recent study outcomes (acelERA BC, EMERALD, SERENA-2, and EMBER-3), this discrepancy can be explained by two factors.

- (1) *ESR1* mutations are poorly prognostic on standard-of-care therapy, but experience improved outcomes on next-generation SERDs. This led to the approval of elacestrant for *ESR1*-mutated disease.
- (2) When mutant and NMD disease are both treated with next-generation SERDs, the difference in prognosis becomes non-significant (as in Fig. 2b, Extended Data Fig. 3a-b).

Using the acelERA BC as an example: the median PFS for *ESR1*-mutant patients receiving standard-of-care was 3.8 months, *ESR1*-mutant patients receiving giredestrant was 5.3 months, and all patients (ITT; includes both *ESR1*-mutant and NMD) was 5.4 and 5.6 in standard-of-care and giredestrant arms. We have now dedicated the second paragraph of the Discussion section to explaining this point. (*During manuscript revisions, we identified a mislabeling in the previous Fig. 4a, which may have added to the confusion; this error has now been corrected.)

REVIEWER COMMENTS

Reviewer #1 (Remarks to the Author):

I find this manuscript compelling for understanding the response and resistance mechanisms associated with SERDs, SERMs, CDK4/6 inhibitors, and the combination of SERD and CDK4/6 inhibitors. The laboratory work and analysis of clinical samples complement each other, creating a strong synergy in the research.

All of the reviewer's comments have been addressed accurately and systematically. Many new experiments and analyses have been incorporated, leading to valuable research findings that have been effectively presented.

We thank the Reviewer for this feedback.

In Figure 3, the response to Tamoxifen (TAM) appears bell-shaped. From a target concentration perspective, it seems that certain regulatory mechanisms may be involved.

This is indeed an interesting observation, and may be linked to differences in mechanism of action between selective ER modulators (tamoxifen) vs. selective ER degraders (fulvestrant and giredestrant). We have now highlighted this observation on line 305 of the main text.

Reviewer #4 - Replacement for Reviewer #3 (Remarks to the Author):

Through the revisions, Liang et al appropriately reframed analyses to account for the difference in prior CDK4/6 inhibition, more clearly demonstrating the differences in responders versus non-responders to giredestrant. Their expanded in vitro analysis across resistant cell lines adequately showed common pattern of loss of ER signaling and an up-regulation of alternative pathways, EGFR/MAPK and Hippo/TEAD. The additional ATAC-seq data exploring a mechanistic underpinnings for endocrine resistance and loss of luminal features. Demonstrating the limitations of mutation calls over ddPCR are clinically important and should be highlighted. Further delineating those patients who had prior CDK4/6i treatment vs those who are treatment naive will continue to improve the applicability of the study population to real-world patients, most of whom will have received CDK4/6 therapy. I have a few additional minor comments below.

We thank the Reviewer for this feedback.

Minor comments:

Line 201, pg6: while definitions of luminal-high and luminal-low is clearly written in the methods, it would benefit the reader to add a brief definition within the main body of the text rather than the figure legend.

We have now included a definition of these terms at the first instance they are mentioned, on line 195 of the main text.

Figure 4e: I could not follow what the two different barplots refer to. What is the difference between the left and right parts of 4e? They look identical with no indicating title or description in the figure legend.

As specified in the x-axis labels, the left and right part of Fig. 4e refer to the two different GiredR cell lines that were profiled by whole exome sequencing: MCF7-GiredR on the left and T47D-GiredR on the right.

Extended Data Figure 2: is it possible to add prior CDK4/6 treatment as a color bar under response? This could aid in visualization and interpretation of the figure.

We thank the Reviewer for this great suggestion, and have updated Extended Data Fig. 2c with prior CDK4/6i treatment information.

Extended Data Figure 3b: for the panel demonstrating Treatment arm, is it possible to also demonstrate the survival curve within the Combo group of those who had received prior CDK4/6 inhibition?

While this would be an interesting analysis in principle, the vast majority of patients in the Combo arm of the study were CDK4/6i-naïve, as illustrated in Extended Data Fig. 3c. Only 5 patients in the Combo arm received prior CDK4/6i. Given the limited number of prior CDK4/6i-exposed Combo arm patients, the proposed analysis would be difficult to interpret.

Reviewer #5 - Breast cancer epigenetics (Remarks to the Author):

In the manuscript “Loss of luminal lineage drives resistance to next-generation ERA-antagonists in pretreated ER+ HER2- locally-advanced or metastatic breast cancer”, Liang et al. investigate the mechanistic basis for response to treatment in the phase 1a/b study with the oral selective estrogen receptor degrader (SERD) giredestrant (NCT03332797) alone or in combination with a CDK4/6 inhibitor. The authors have performed exploratory analyses of both tumor biopsies and ctDNA from patients on the trial and in addition have developed in vitro cell line models of giredestrant resistance. In the trial patients with metastatic ER+/HER2- breast cancer were treated with giredestrant at several different dose levels and a subset were treated with the combination of giredestrant plus palbociclib. Given the nature of the trial and the lack of a control arm, the results are largely hypothesis generating. Nonetheless the data from the trial are valuable and should be reported.

We thank the Review for providing thoughtful feedback, which has greatly approved the manuscript. As described below, we have added new data (Figure 4f-g, new Extended Data Figure 9, and Extended Data Figure 10c-d) to address the Reviewer’s comments. We have also edited the relevant manuscript text to clarify key points or describe the new data. For the Reviewer’s convenience: in the revised manuscript, text which has been added/edited since the previous submission is colored in **red** while unaltered text is in **black**.

Major critiques:

1) While the primary data from the in vitro models has been deposited in GEO, the RNA-seq and liquid biopsy data from the trial should also be made available.

We wholeheartedly agree with the importance of data sharing in research and make our data publicly available whenever possible. Unfortunately for this clinical trial, we do not have the patients’ consent to publicly share their sequencing results with the scientific community. Prior to participating in our clinical trial, patients signed an informed consent form, which stipulated the terms under which their data could be used. Two key terms of data usage are: (1) our organization is only allowed to use the patients’ data for a fixed amount of time (duration varies by country laws), after which we are legally/ethically required to delete the data and (2) patients have the additional right to withdraw their consent for us to use their data at any time (some patients still remain on this study). This means that we can publish any analyses we performed prior to the time limit and/or consent withdrawal, but we cannot make individual patients’ sequencing data publicly available in perpetuity.

As an alternative, we will provide all sequencing-derived values used in this manuscript (e.g. per patient ER activity scores, luminal TF scores, etc.) as source data, which can still be of great benefit to the community.

2) The authors argue that in the giredestrant resistant models that the inability to identify a clear genetic mechanism of resistance suggests that resistance is primarily epigenetically driven. The data presented do not support this conclusion. The stable resistance phenotype of GirdR models may indicate genetic selection and there may be different genetic mechanisms in MCF7 or T47D cells. In addition, there may be multiple different genetic mechanisms in distinct clones present in the GirdR cultures that may mask

distinct genetic resistance mechanisms. Single cell cloning or sequencing would be needed to address this possibility. In Figure 3e, using whole exome sequencing (WES), over 50 mutations are enriched in GirdR cells, including known cancer-associated variants, but follow-up experiments are not performed. Additionally, copy number variation (CNV) analysis was performed on GirdR cells, which identified a loss on chromosome X including 19 cancer associated genes (Line 335), but this data is not presented in the figure and there is no validation. Deletions found in the CNV analysis further supports genetic selection of a SERD resistant clone. The authors state “Since few genetic mutations were acquired by GiredR cells, we considered whether epigenetic alterations may drive ER α independence and loss of luminal identity” (Line 362), and in Figure 5 they propose an epigenetic model of endocrine resistance.

The authors should point out more clearly that genetic mechanisms of resistance remain a likely explanation despite their inability to identify them.

We agree with the Reviewer’s assessment, and have made multiple changes to address this point. We underscore the fact that our data does not preclude genetic mechanisms of resistance as follows:

- We now conclude the whole exome sequencing results paragraph by explicitly stating that “genetic drivers for giredestrant resistance remain a possibility.” (line 343).
- At the start of the following section, we have replaced the first sentence (line 362 in the previous version, now line 364) with “We considered whether epigenetic alterations may promote ER α independence and loss of luminal identity upon prolonged ER α inhibition,” to avoid the implication that genetic vs. epigenetic mechanisms of resistance are mutually exclusive.
- For the cartoon model in Figure 5, we have revised the title into a descriptive statement: “The shifting epigenetic landscape in ER α -independent, giredestrant-resistant disease”.

3) This manuscript relies heavily on computational imputation without validation. For example, in Figure 4d-f, ATACseq identified chromatin opening at FOXM1 motifs, which the authors argue is a novel driver of endocrine resistance (Line 34), however the ATACseq results do not necessarily demonstrate a functional role for FOXM1. For example, the FOXM1 motif highly resembles the FOXA1 motif, a known driver of endocrine resistance. A direct role for FOXM1 should be validated using FOXM1 ChIPseq and testing whether FOXM1 is necessary for resistance in GirdR cells and sufficient to induce resistance in parental cells.

To directly test the roles of FOXA1 and FOXM1, we have performed ChIP-seq against both factors in parental vs. GiredR cells for MCF7 and T47D models. The new ChIP-seq data confirm differential binding of both FOXA1 and FOXM1 in GiredR cells, and corroborate the prior ATAC-seq findings. FOXM1 is enriched at 823 and 3,222 sites in GiredR cells vs. parental, for the respective models. The data are presented in the Figure 4f-g and Extended Data Figure 9 of the revised manuscript, and described on lines 414 – 432. Raw ChIP-seq data has been uploaded to the GEO under GSE305432.

We also explored genetic perturbations of *FOXM1*. Overexpression of *FOXM1* in MCF7 cells was previously linked to endocrine resistance, and drove increased aggressiveness and metastatic potential (Yang et al. 2013, Bergamaschi, et al. 2014). We therefore focused on the effect of *FOXM1* siRNA in parental and GiredR cells (new Extended Data Fig. 10c-d). *FOXM1* siRNA in parental cells suppressed cell growth by Incucyte measurement, consistent with a prior report (Yang et al. 2013). However, *FOXM1* siRNA did not alter the growth of either GiredR model. Together, the data suggests that although FOXM1 has an expanded range of chromatin binding in the GiredR state, its knockdown does not affect growth rate in GiredR cells.

Given these data, we have refrained from describing FOXM1 as a “driver” of resistance and removed the statement in the Discussion section that speculated on the “potential for FOXM1 as a therapeutic target”. We have also de-emphasized the FOXM1 results by moving them to Extended Data; Figure 4 now focuses on FOXA1, which the Reviewer highlights as a known driver of endocrine resistance. Nonetheless, our single siRNA experiment does not preclude other functional roles for FOXM1, such as

establishment of resistance over the course of treatment. Further experiments on FOXM1 may remain worthwhile – for example, it would be interesting to test whether *FOXM1* knockdown affects the rate at which parental cells acquire giredestrant resistance when exposed to long-term treatment.

4) The use of the FOXM1 inhibitor FDI-6 is not convincing. What is the evidence that its effects are on-target? Can the effects be reproduced genetically?

We have removed the FDI-6 data from the manuscript along with associated text, and instead explored genetic perturbation of *FOXM1*. Those results are described in our response to point #3 above.

5) The model presented in Figure 5 is not supported by the data presented.

We have now modified the model to capture key conclusions supported by our data. We feel this revised model is supported by either (1) data which are newly-presented in this manuscript or (2) are established concepts in the field. Specific changes include:

- FOXA1 and FOXM1 are illustrated as binding to the ER-independent, GiredR-induced sites, as supported by our new ChIP-seq data.
- We have removed any additional transcription factors from the GiredR-induced sites (e.g. TEAD), since their previous inclusion was only based on ATAC-seq motif enrichment analyses.
- We now clearly distinguish conclusions from the patient vs. cell line data by dividing the model's layout and including labels specifying the transition from parental to GiredR states *in vitro*.
- In endocrine-sensitive cells, ER is depicted as associating with FOXA1 and FOXM1 to regulate ER-associated chromatin sites. While we do not perform ER ChIP-seq in our study, we feel this portion of the model is firmly supported by multiple prior publications (Hurtado et al. 2011, Sanders et al. 2013, and others).

Other questions and minor concerns:

1) Are any *ESR1* gene amplifications, deletions, or gene fusions identified in patients?

We did not detect any *ESR1* copy number alterations (amplifications and deletions) among the $n = 123$ patient samples tested on our study. Gene fusions involving *ESR1* were also not detected.

2) The methods section indicates that WES was performed using a mouse genome kit (Line 1045).

Apologies for this oversight - the WES methods language was copied from a prior manuscript in which WES was performed on mouse tissues. We have now corrected the corresponding methods section to reflect the SureSelect Human V7 kit, which was used for these experiments.

3) Please define NMD (Line 128) and specify the genes for the Luminal TF (Figure 1F) and the MAPK signature (Ext Data Fig 9a) in this manuscript, and update Ref #26.

We have now defined NMD (no mutation detected) on line 128, and included the genes comprising the requested signatures in the Methods, under the "Gene signature scores" sub-section of "Bioinformatics data analysis". Reference 26 has also been updated.

Point-by-point Responses to Reviewers

Below, please find our responses to the Reviewer's comments; the Reviewer's comments are colored blue and our responses are in black.

REVIEWERS' COMMENTS

Reviewer #4 (Remarks to the Author):

I have no further comments for the authors. The prior comments raised have all been adequately addressed, and the manuscript should be shared with the broader community.

We thank the Reviewer for providing thoughtful feedback, which has greatly improved our manuscript.

Reviewer #6 (Remarks to the Author):

This study addresses an important clinical issue: the variable response to next-generation SERDs in advanced ER⁺/HER2⁻ breast cancer. Using multi-omics data from a giredestrant clinical trial and in vitro resistance models, the authors conclude that loss of luminal lineage identity, rather than ESR1 mutations, underlies resistance to ER α -targeted therapy. The study is strong, methodologically rigorous, and offers meaningful mechanistic and clinical insights. However, some key concerns are needed to be addressed.

Major Comments:

1. The authors provided strong evidence for the role of FOXA1 in driving a resistant phenotype through chromatin remodeling. However, the mechanism by which FOXA1 expression and activity are upregulated in the resistant cells remains unclear. Is this a stochastic event, or is it driven by a specific upstream signaling pathway? Furthermore, the authors show that ~70% of FOXA1 binding sites in resistant cells are not co-occupied by ER α . What are the transcriptional partners of FOXA1 at these sites, and what are the key target genes that drive the resistant phenotype? Have the authors considered performing ChIP-seq for FOXA1 in their resistant cell lines to identify its direct targets?

ChIP-seq against FOXA1 was performed in both MCF7 and T47D parental and GiredR cell lines (Figure 4f-g, Supplementary Fig. 9b-d). Together, the data support altered FOXA1 binding in parental vs. GiredR cell lines. In GiredR cells, FOXA1 is bound to chromatin sites containing TEAD1/4 motifs, AR motifs, and NFI-family motifs. Regarding the "upregulation of FOXA1 activity" following long-term giredestrant treatment, we posit that the absence of functional ER α frees FOXA1 to bind/regulate ER-independent chromatin regions.

2. The study focuses on a specific mechanism of resistance involving the loss of luminal lineage. However, it is likely that other mechanisms of resistance also exist. The authors themselves show that a subset of non-responders still retain some ER α activity. How do the authors envision the interplay between the mechanism they have identified and other known resistance mechanisms, such as the acquisition of ESR1 mutations or the activation of other receptor tyrosine kinases? A more nuanced discussion of the heterogeneity of resistance would strengthen the manuscript.

This is an excellent point, and we now address this in the Discussion section of our manuscript. Multiple mechanisms are likely required to drive resistance to giredestrant. In particular, the primary effect of luminal lineage loss is to abrogate ER dependency; this is insufficient to enable sustained tumor proliferation. Instead, other pathways such as MAPK/RTK or Hippo/TEAD signaling are needed to sustain tumor cells.

3. The manuscript demonstrates upregulation of multiple bypass pathways (EGFR/MAPK, Hippo/TEAD, Wnt, IL2/STAT5) in non-responder tumors. However, it remains unclear whether these pathway activations are drivers of resistance or merely consequences of losing ER α -dependent growth control. Have the authors performed functional validation experiments (e.g., genetic or pharmacological inhibition of these pathways) to determine whether blocking these pathways can restore giredestrant sensitivity?

In Figure 4i, we tested whether GiredR cells acquired sensitivity to MEK (MAPK pathway) and TEAD inhibition. GiredR cells were more sensitive to growth inhibition via MEKi or TEADi than parental cells, although the effect was more pronounced in MCF7-GiredR than in T47D-GiredR. Thus far, we have not tested inhibitors of Wnt or STAT signaling; per feedback from the *Nature Communications* editors, additional experiments are beyond the scope of this manuscript.

4. The observation that different resistant cell lines (MCF7-GiredR vs. T47D-GiredR) show differential activation of bypass pathways (MAPK in MCF7 but not T47D) suggests context-dependent resistance mechanisms. What determines which bypass pathway becomes activated in a given cellular context? Are there pre-existing differences in signaling pathway wiring between MCF7 and T47D cells that predispose them to different resistance mechanisms?

This is a very interesting question - we have added a discussion of this point in the Discussion section of our manuscript. It is highly plausible that baseline differences in genetic background and epigenetic state across cell lines would predispose each cell line towards acquiring different bypass mechanisms.

5. The authors identify TEAD4 and TCF4 as transcription factors associated with non-responder genes, implicating Hippo/TEAD and Wnt signaling. However, canonical activation of these pathways typically involves nuclear translocation of YAP/TAZ (for Hippo) or β -catenin (for Wnt). Have the authors examined the subcellular localization and activation status of these key signaling mediators in resistant cells? Are there genetic alterations in pathway components that would explain constitutive activation?

Based on the mutational profiling performed on patients, we did not observe genetic alterations in TEAD or Wnt pathway components which could explain the increased pathway activity in NR patients (Supplementary Fig. 2c). Thus far, we have not examined the localization or activity status of TEAD or Wnt in GiredR cells; per feedback from the *Nature Communications* editors, additional experiments are beyond the scope of this manuscript.

6. The authors also generated resistance models using GDC-0810 (a SERD-SERM hybrid) and observed similar features. Have they tested other next-generation SERDs (e.g., elacestrant, camizestrant) to determine whether this resistance mechanism is common to all SERDs or specific to certain chemical scaffolds?

We have not generated resistant cell lines from other next-generation SERDs. However, based on the known pharmacology of other SERDs like elacestrant or camizestrant, we predict that this resistance mechanism would apply to both oral SERDs and SERD-SERM hybrids, since loss of ER activity/dependence would fundamentally render breast cancer cells insensitive to any ER-targeted therapy. Per feedback from the *Nature Communications* editors, additional experiments are beyond the scope of this manuscript.

Minor Comments:

1. In Figure 2, the authors show that several transcription factors are associated with non-responder genes. It would be helpful to show the expression levels of these TFs in the responder versus non-responder patient samples.

Per feedback from the *Nature Communications* editors, additional data are beyond the scope of this manuscript revision.

2. The authors mention that the MCF7-GiredR cells have deletions on chromosome X. Are any of the deleted genes known to be involved in epigenetic regulation or ER signaling? A more detailed discussion of the potential functional consequences of these deletions would be welcome.

Several deleted chrX genes (including ATP6AP1) have been linked ER activity. We will expand our discussion in the corresponding results section of the manuscript text.

3. The sample sizes for some comparisons are relatively small (e.g., n=10 NR vs. n=17 Resp pairs for the on-treatment analysis). Have the authors performed power calculations to determine whether they have sufficient statistical power to detect meaningful differences?

The paired analyses (Supplementary Fig. 4a) were limited due to difficulties in collecting matching baseline and on-treatment specimens in our clinical trial. In a power calculation (assuming 80% power and $\alpha = 0.05$), our minimum detectable effect size was $d = 1.15$.

4. Ensure consistent use of drug-class terminology (e.g., “next-generation ER α -antagonists,”

“SERD-like,” “PROTAC-like”), and consistently name or classify the compound used throughout the manuscript.

We have revised our manuscript to ensure consistent use of drug-class terminology.

Reviewer #7 (Remarks to the Author):

We thank the Reviewer for contributing to the review process.